# State transitions in the substantia nigra reticulata predict the onset of motor deficits in models of progressive dopamine depletion in mice

Amanda M Willard[1,2†], Brian R Isett[1†], Timothy C Whalen[2], Kevin J Mastro[3], Chris S Ki[4], Xiaobo Mao[5,6], Aryn H Gittis[1,2]*

[1]Department of Biological Sciences, Carnegie Mellon University, Pittsburgh, United States; [2]Center for the Neural Basis of Cognition, Carnegie Mellon University, Pittsburgh, United States; [3]Boston Children's Hospital and Harvard Medical School, Boston, United States; [4]University of California, Berkeley, Berkeley, United States; [5]Neuroregeneration and Stem Cell Programs, Institute for Cell Engineering, Johns Hopkins University School of Medicine, Baltimore, United States; [6]Department of Neurology, Johns Hopkins University School of Medicine, Baltimore, United States

*For correspondence:
agittis@cmu.edu

†These authors contributed equally to this work

Competing interests: The authors declare that no competing interests exist.

**Abstract** Parkinson's disease (PD) is a progressive neurodegenerative disorder whose cardinal motor symptoms are attributed to dysfunction of basal ganglia circuits under conditions of low dopamine. Despite well-established physiological criteria to define basal ganglia dysfunction, correlations between individual parameters and motor symptoms are often weak, challenging their predictive validity and causal contributions to behavior. One limitation is that basal ganglia pathophysiology is studied only at end-stages of depletion, leaving an impoverished understanding of when deficits emerge and how they evolve over the course of depletion. In this study, we use toxin- and neurodegeneration-induced mouse models of dopamine depletion to establish the physiological trajectory by which the substantia nigra reticulata (SNr) transitions from the healthy to the diseased state. We find that physiological progression in the SNr proceeds in discrete state transitions that are highly stereotyped across models and correlate well with the prodromal and symptomatic stages of behavior.
DOI: https://doi.org/10.7554/eLife.42746.001

## Introduction

Parkinson's disease (PD) is a movement disorder caused by the progressive degeneration of dopamine neurons in the substantia nigra pars compacta (SNc). The main projection targets of SNc dopamine neurons are motor territories of the basal ganglia, especially the dorsal striatum (*Björklund and Dunnett, 2007*; *Lavoie et al., 1989*). As a result, the classical symptoms of PD are motor, including resting tremor, postural abnormalities, gait disturbances, and decreased/slowed movement. Physiological indicators of basal ganglia dysfunction in the parkinsonian state include changes in both neuronal firing rates and patterns (*Bevan et al., 2002*; *Hammond et al., 2007a*; *Obeso et al., 2000*; *Wichmann and Dostrovsky, 2011*).

The 'rate model' posits that motor symptoms are the result of elevated firing rates of basal ganglia output neurons under dopamine depleted conditions (*Albin et al., 1989*; *DeLong, 1990*). However, changes in both the magnitude and sign of firing rates vary widely across studies, suggesting that rates alone do not account entirely for motor deficits (*Leblois et al., 2007*; *Nelson and Kreitzer, 2014*; *Seeger-Armbruster and von Ameln-Mayerhofer, 2013b*; *Wichmann et al., 1999*). An

alternative model is that motor deficits are more closely related to changes in neuronal firing patterns (*Guridi and Alegre, 2017*; *Hammond et al., 2007a*). Under dopamine depleted conditions, basal ganglia output neurons fire more irregularly and synchronously than under normal conditions (*Filion and Tremblay, 1991*; *Galvan and Wichmann, 2008*; *Heimer et al., 2002*; *Hutchison et al., 1994*; *Wichmann et al., 2002*), but the relationship between firing patterns and the manifestation of motor symptoms is complex (*Muralidharan et al., 2016*; *Sanders et al., 2013*; *Wichmann and Soares, 2006*; *Wichmann et al., 1999*).

Despite a vast literature describing basal ganglia pathophysiology at end-stages of dopamine loss, the question of *when* deficits emerge over the course of progressive depletion is poorly understood. PD is a gradual, neurodegenerative disorder and motor symptoms rarely present until late stages of dopamine loss (~20–30% striatal dopamine remaining) (*Bernheimer et al., 1973*; *Fahn, 2003*; *Riederer and Wuketich, 1976*). At this stage, treatments are limited to those that minimize symptoms rather than disease-modifying therapies. Diagnosing and treating patients during the presymptomatic, or 'prodromal' phase of the disease – when dopamine levels have started to decline but motor symptoms are not yet present – would increase options for treatment and result in better patient outcomes (*Olanow and Obeso, 2012*; *Schapira and Tolosa, 2010*; *Tolosa et al., 2009*). However, little is known about the physiological trajectory of basal ganglia circuits as they transition from the healthy to the diseased state. Do physiological changes emerge all at once, or is there a hierarchical progression? Does their severity worsen monotonically, or are there discrete inflection points that predict the transition from asymptomatic to symptomatic stages of the disease? Answering these questions is critical to develop therapeutic strategies for early intervention.

To study the onset and progression of basal ganglia pathophysiology during progressive dopamine loss, we recorded from the substantia nigra pars reticulata (SNr) of mice at different severities of dopamine depletion, induced at rates ranging from 3 days – 6 months, using both toxin and neurodegenerative models. We found that across models, SNr pathophysiology progressed as transitions through discrete physiological states that were highly stereotyped across models and independent of the rate or mechanism of depletion. Our results suggest that basal ganglia output is initially very sensitive to dopamine loss and small decreases in dopamine are sufficient to transition the SNr from its normal physiological state into a 'moderate' pathophysiological state. In the moderate pathophysiological state, the SNr is robust to continued decreases in dopamine levels, possibly reflecting compensatory plasticity, until undergoing a final state transition into the 'severe' pathophysiological state when dopamine levels drop below 25–35% remaining. These results reveal key inflection points in the progression of basal ganglia pathophysiology that correlate with symptomatic manifestation over the course of progressive dopamine loss.

## Results

### SNr pathophysiology at End-Stages of dopamine depletion in awake mice

To establish the electrophysiological parameters that define SNr pathophysiology at end-stages of dopamine loss, we performed in vivo recordings from the SNr of awake, bilaterally depleted mice (*Figure 1A–C*). Recordings targeted to the SNr were confirmed by the presence of fast, tonically active units (~30–40% were modulated by movement), and by visualizing the recording track postmortem with immunostaining against the microglial marker, Iba-1 (*Figure 1B*). Depletions were induced by infusing 6-hydroxydopamine (6-OHDA) into the bilateral medial forebrain bundle (MFB), either in a single, high-dose infusion ('acute';~3 days), or through a series of repeated, low-dose infusions, spaced 5 days apart ('gradual'; 38 ± 15 days). In all mice, depletion severity was quantified using tyrosine hydroxylase (TH) immunoreactivity in the striatum, a metric that is well correlated with tissue dopamine levels (*Willard et al., 2015*). On average, striatal TH levels were 4.1 ± 2.6% (relative to control) in acutely depleted mice and 3.7 ± 4.4% in gradually depleted mice. Because mice at advanced stages of depletion rarely initiated movements on the wheel (<2% of time spent moving), SNr physiology across conditions was compared during periods of rest (see: Materials and methods).

In dopamine depleted mice, SNr firing rates were significantly lower than those measured in dopamine intact controls (*Figure 1D*) (Median ±MAD: Ctl: 35 ± 11 Hz, *n* = 262 neurons/7 animals; Acute: 25 ± 11 Hz, *n* = 245 neurons/7 animals; Gradual: 26 ± 11 Hz, *n* = 293 neurons/9 animals; KW

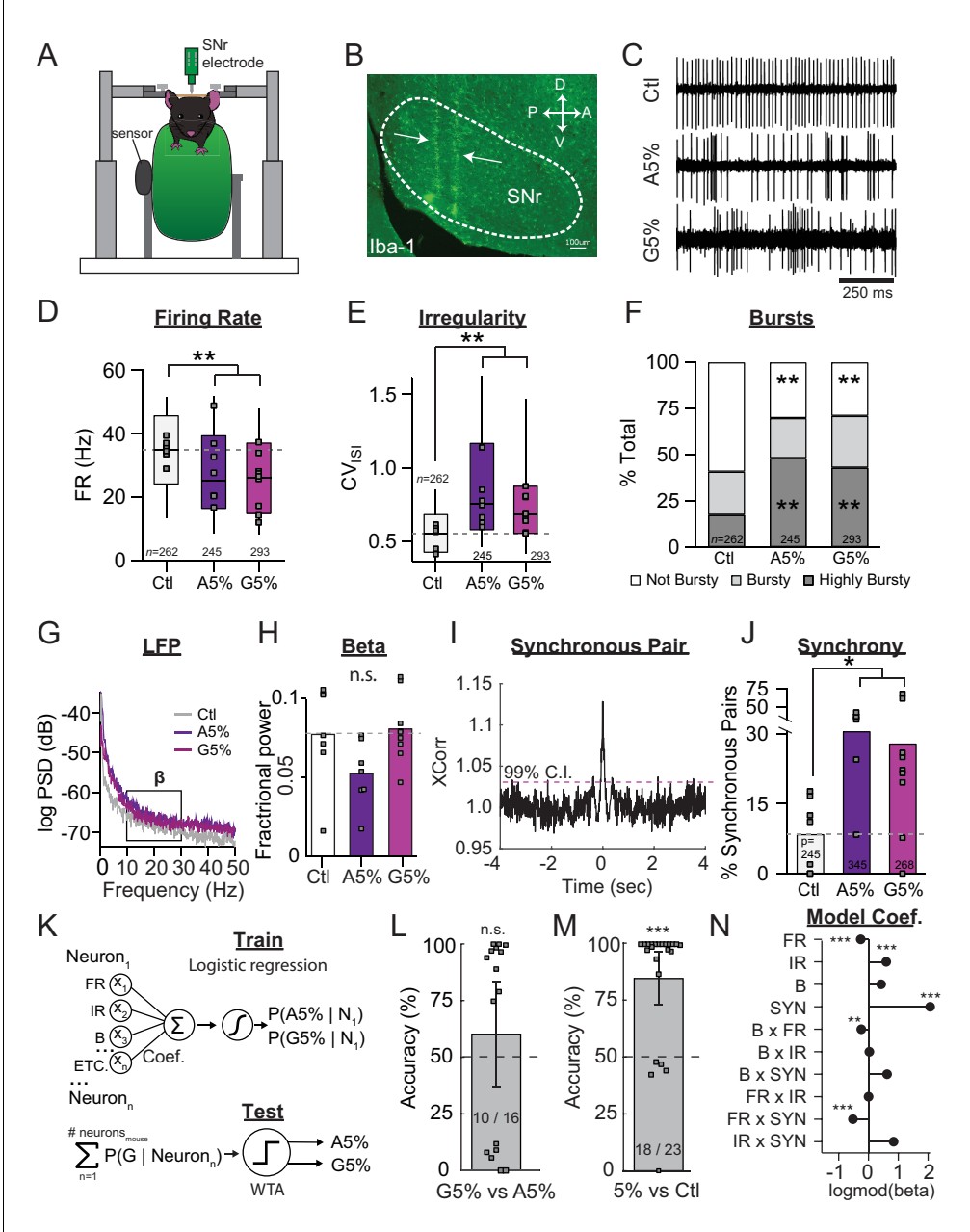

**Figure 1.** SNr pathophysiology at late stages of dopamine depletion is similar between acutely and gradually depleted mice. (**A**) Schematic of in vivo recording setup depicting head-restrained mouse on top of a freely moving wheel with a sensor to record wheel movement and a linear silicone probe to record SNr units. (**B**) Sagittal section showing recording locations, revealed with an immunostain against Iba-1. Scale Bar = 100 μm. (**C**) Representative examples of single-units in the SNr recorded from control, end-stage acute (A5%), or end-stage gradual conditions (G5%). (**D**) Box plot of firing rates of all single units recorded in each condition. N's denote number of single units in each condition. Grey squares indicate medians for individual mice. KW, $\chi^2(2) = 40.463$, p < 0.0001, pairwise, **p < 0.005 from Ctl. (**E**) Box plot of $CV_{ISI}$ of all single units recorded in each condition. KW, $\chi^2(2) = 101.830$, p < 0.0001, pairwise, **p < 0.005. (**F**) Proportion of 'not bursty,' 'bursty,' and 'highly bursty' units in each condition (see Materials and methods). Pearson, $\chi^2(4) = 80.591$, p < 0.0001, z-test, **p < 0 .005 from Ctl. (**G**) Representative LFP spectrograms from Ctl, A5%, and G5% with box highlighting β frequency range (13–30 Hz). (**H**) Mean fractional β power (power in β relative to power 1–100 Hz) at each stage of depletion. Grey squares indicate animal means. ANOVA, p = 0.076. (**I**) Example cross-correlogram showing synchronous spiking between a pair of simultaneously recorded SNr units. Horizontal dotted line is 99% confidence interval. (**J**) Proportion of synchronous pairs in each condition (calculated as a percentage of total pairs); grey squares indicate proportion of synchronous

*Figure 1 continued on next page*

*Figure 1 continued*

pairs calculated per animal. ANOVA, F(2) = 3.992, p = 0.035, Dunnett, *p < 0.05 from Ctl. (**K**) *Top:* Classifier trained to predict the probability that a neuron belongs to A5%, G5% groups using: FR, firing rate; IR, irregularity; B, percent spikes in bursts; synchrony and all pair-wise multiplicative interactions ('ETC.'). *Bottom:* predicted probabilities of A5% and G5% group membership were summed across neurons from each held-out mouse (jack-knife), and thresholded using a Winner-Take-All criterion (WTA). See: Materials and methods. (**L**) Mean cross-validated accuracy of predicting which depletion model led to endstage in each mouse (500 permutations; $\pm CI_{95}$). Right-tail t-test vs. chance, 50%, p = 0.3842, n = 16 mice. Grey squares indicate mean prediction for each mouse. Inset text indicates mice predicted >chance/ total mice. (**M**) Same as J but predicting A5% and G5% combined (5%) vs Ctl. Right-tail t-test vs. chance, 50%, p = $7.87 \times 10^{-6}$, n = 23 mice. (**N**) Coefficients from average model in *M* describing successful 5% vs. Ctl discrimination. Significance by $CI_{95}$. *p < 0.05, **p < 0.01, ***p < 0.001.

DOI: https://doi.org/10.7554/eLife.42746.002

pairwise, Ctl vs. Acute, Grad, p < 0.0001). Firing patterns of SNr neurons were also significantly changed by dopamine depletions. Firing was more irregular in depleted mice, quantified as a significant increase in the coefficient of variation of the interspike interval ($CV_{ISI}$) (**Figure 1E**) (Median $\pm$MAD: Ctl: 0.54 $\pm$ 0.13; Acute: 0.76 $\pm$ 0.24; Gradual: 0.68 $\pm$ 0.16; KW pairwise, Ctl vs. Acute, Grad, p < 0.0001). Burst firing also became more prevalent in dopamine depleted mice, quantified using the Poisson Surprise method (*Soares et al., 2004*) (**Figure 1F** and see Materials and methods) (z-test, Ctl vs. Acute, Grad, p < 0.001).

To assess physiological changes at the population level, we first inspected the spectral power of local field potentials (LFPs) (**Figure 1G**). Increased spectral power in the beta frequency range (13–30 Hz, 'β−oscillations') has been observed in human PD patients (*Brown et al., 2001*; *Hammond et al., 2007b*; *Jenkinson and Brown, 2011*; *Kühn et al., 2006*; *Quinn et al., 2015*) and some animal models of dopamine depletion. To quantify fractional β power, we divided the power in the β range by total power in 1–100 Hz of the local field potential (**Figure 1H**). We observed no difference in fractional β between Ctl, Acute and Grad mice (1-way ANOVA, p = 0.076), consistent with a previous report (*Lobb et al., 2013*). These results suggest that elevated β-oscillations are not a robust feature of SNr pathophysiology in 6-OHDA depleted mice.

To examine more directly whether spiking synchrony is affected by dopamine depletion, we calculated the cross-correlation of pairs of spike trains over short time windows and normalized each window to account for nonstationarities in firing rates over a recording session. We then calculated a 99% confidence interval for each pair. A pair was termed 'synchronous' if its correlation at zero lag exceeded this confidence interval (**Figure 1I**, see Materials and methods). We found that the average percentage of synchronous pairs was 8.6 $\pm$ 7.8% (*n* = 7 animals) in control mice but increased to 30.6 $\pm$ 11.4% (*n* = 7 animals) and 28.0 $\pm$ 22.6% (*n* = 9 animals) in acutely and gradually depleted mice, respectively (**Figure 1J**) (Dunnett, Ctl vs. Acute, Grad, p < 0.05).

Taken together, our results confirm the presence of multiple physiological changes in SNr neurons under dopamine depleted conditions. Although no single parameter appeared to reliably distinguish gradually from acutely depleted mice, it is possible that simultaneous changes in multiple parameters distinguish these two conditions. To test this possibility, we trained a classifier to discriminate acutely from gradually depleted mice using the pathophysiological metrics described above (**Figure 1K**, and see Materials and methods). The classifier was unable to discriminate these conditions above chance levels (**Figure 1L**) (60.2 $\pm$ 24% $CI_{95}$, right-tail t-test vs. chance, 50%, p = 0.3842, n = 16 mice). By contrast, retraining the same model to discriminate end-stage mice (Grad +Acute) from control mice resulted in classification well above chance (**Figure 1M**) (84.9 $\pm$ 12% $CI_{95}$, right-tail t-test vs. chance, 50%, p = $7.87 \times 10^{-6}$, n = 23 mice). Importantly, the majority of mice from both depletion models were correctly classified (G5% = 7/9 mice, A5% = 6/7 mice). Examination of the model coefficients showed that successful discrimination largely depended on the pathophysiological changes described above, as well as interactions between FR and spike patterns (**Figure 1N**). These results suggest that stereotyped changes in SNr physiology occur at end-stages of dopamine loss, regardless of whether dopamine is depleted slowly over a month, or acutely, over days.

# SNr pathophysiology proceeds in two phases during gradual dopamine depletion

To determine how SNr pathophysiology develops over the course of gradual dopamine loss, we performed in vivo recordings from mice at different stages of depletion (*Figure 2A–B*). Because precise quantification of depletion required postmortem analysis, data at each stage were collected in different groups of mice (TH immunoreactivity relative to control): 'G85%'=86 ± 9%; 'G60%'=61 ± 12%; 'G30%'=28 ± 18%; 'G5%'=3.7 ± 4.4%.

Even at early stages of dopamine depletion, SNr physiology showed differences relative to control. The most sensitive parameter was firing rate (*Figure 2C*), which was significantly reduced in the earliest depletion group examined (G85%) and remained suppressed as depletions progressed (Median ±MAD: Ctl: 35 ± 11 Hz, $n$ = 262 neurons/7 animals; G85%: 26 ± 12 Hz, $n$ = 272/5; G60%: 27 ± 14 Hz, $n$ = 318/7; G30%: 30 ± 10 Hz, $n$ = 318/8; G5%: 26 ± 11 Hz, $n$ = 293/9; KW pairwise, Ctl vs. 85%, 60%, 30%, 5%, $p < 0.001$).

Changes in firing patterns were more pronounced at later depletion stages. Firing irregularity (CV$_{isi}$) in the G85% group was modestly, but significantly elevated relative to control (*Figure 2D*) but this difference became more pronounced at later depletion stages, with <60% dopamine remaining (*Figure 2D*) (Median ±MAD: Ctl: 0.54 ± 0.13; G85%: 0.59 ± 0.15; G60%: 0.65 ± 0.20; G30%: 0.71 ± 0.22; G5%: 0.68 ± 0.16; KW pairwise, G85% vs. G30%, G5%, $p < 0.01$). Similarly, the

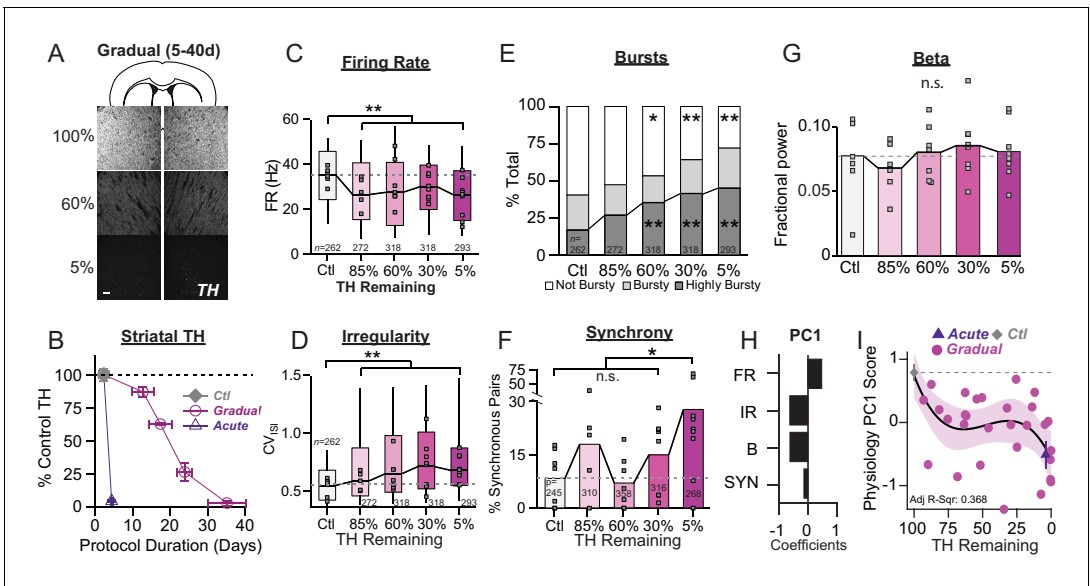

**Figure 2.** Onset and progression of SNr pathophysiology during gradual dopamine depletion with 6-OHDA. (**A**) Coronal sections showing representative images of TH immunoreactivity (TH-IR) in the dorsal striatum of mice treated with saline or bilateral injections of 6-OHDA ('60%' after three injections or '5%' after six injections). Scale Bar = 100 µm. (**B**) Graph showing the trajectory of dopamine depletion induced by 6-OHDA infusions in acutely vs. gradually depleted mice. Percent TH remaining (relative to saline controls) is graphed as a function of protocol duration, measured from the day of first 6-OHDA infusion to day of sacrifice following in vivo recordings. (**C**) Box plot of firing rates of all neurons recorded at each stage of depletion. Grey squares indicate animal medians. KW, $\chi^2(4)$ = 41.846, $p < 0.0001$, pairwise, **$p < 0.005$ from Ctl. (**D**) Box plot of CV$_{ISI}$ recorded for all neurons at each stage of depletion. Grey squares indicate animal medians. KW, $\chi^2(4)$ = 72.039, $p < 0.0001$, pairwise, **$p < 0.005$ from Ctl. (**E**) Proportion of 'not bursty,' 'bursty,' and 'highly bursty' units at each stage of depletion. Pearson, $\chi^2(8)$ = 78.856, $p < 0.0001$, z-test, *$p < 0.05$ and **$p < 0.005$ from Ctl. (**F**) Proportion of synchronous pairs at each stage of depletion (calculated as a percentage of total pairs sampled at that stage); grey squares indicate proportion of synchronous pairs for each mouse. ANOVA, F(4) = 2.753, $p = 0.045$, Dunnett, *$p < 0.05$ from Ctl. (**G**) Mean fractional β power (power in β relative to power 1–100 Hz) at each stage of depletion. Grey squares indicate animal means. ANOVA, $p = 0.7557$. (**H**) 1st principal component coefficients from PCA performed on Ctl, Gradual and Acute single unit physiology: FR, unit firing rate; IR, unit irregularity (CV$_{ISI}$); B, unit percent spikes in bursts; SYN, percent synchronous pairs per mouse (see Materials and methods). (**I**) Mean animal physiology PC1 scores as a function of dopamine loss with model fit to Ctl and Gradual animals. Shaded region indicates CI$_{95}$ of fit (see Materials and methods); Avg ± SEM for Ctl and Acute indicated.

DOI: https://doi.org/10.7554/eLife.42746.003

proportion of 'highly bursty' neurons was relatively unchanged at early stages of depletion, but became pronounced with <60% dopamine remaining (*Figure 2E*) (z-test, Ctl vs. 60%, 30%, 5%, p < 0.001).

Spike synchrony at each stage of depletion was calculated as described in *Figure 1*. The fraction of pairs firing synchronously in each animal tended to increase at intermediate stages of depletion, but did not reach statistical significance until end-stage (*Figure 2F*) (Ctl: 8.6 ± 7.8% *n* = 7 animals; G85%: 18 ± 14% *n* = 5 animals; G60%: 7.3 ± 7.3% *n* = 7 animals; G30%: 15 ± 11% *n* = 8 animals; G5%: 28 ± 23% *n* = 9 animals; Dunnett, Ctl vs. 5%, p = 0.04). Similar to control and end-stage mice, gradually depleted mice showed no difference in fractional β power (*Figure 2G*; 1-way ANOVA, p = 0.7557).

To determine whether capturing changes across all of these parameters simultaneously would reveal different stages of pathophysiological progression, we performed principal component (PC) analysis (see Materials and methods) (*Figure 2H–I*). Physiology PC1 explained 52.6% of neuronal variability. This component captured variation in firing rate that differed in sign to variation in irregularity, bursting and synchrony (*Figure 2H*). To visualize the trajectory of this component over the full spectrum of dopamine depletion, we plotted each mouse's physiology PC1 score as a function of its striatal TH level (*Figure 2I*). We observed that the physiological transition from control to fully depleted mice followed a biphasic decline (*Figure 2I*). The first transition occurred early in the depletion process, reflecting an acute sensitivity of SNr physiology to small changes in dopamine levels. After this initial drop, physiology PC1 remained relatively stable across a wide range of dopamine levels but underwent a second sharp drop as mice reached the end-stage of depletion. At end-stage, the model fit to gradually depleted mouse physiology PC1 scores overlapped almost identically with the mean physiology PC1 score calculated from acutely depleted mice (*Figure 2I*, 'Acute'). Thus, over the course of 6-OHDA dopamine depletion, mice show early changes in SNr pathophysiology, entering an intermediate period of stable physiology, followed by a final decent to end-stage pathophysiology.

## Gradual dopamine depletion with 6-OHDA results in late onset of behavioral deficits

To investigate the relationship between SNr pathophysiology and symptomatic onset over the course of gradual dopamine depletion, mice were given a battery of behavioral tests prior to in vivo recordings on the same day. Behavioral performance on individual tasks followed a variety of trends in relation to dopamine loss (*Figure 3*). Early dopamine loss often resulted in modest hyperactivity relative to control mice, followed by hypoactivity at late depletion stages, as observed in open field velocity (*Figure 3A*). Velocity, rearing, and total time taken to complete the pole task were preserved until late stages of dopamine loss (*Figure 3B–D*). Wire hang showed a high degree of variability in performance at early stages of dopamine loss that sharply declined when dopamine levels dropped below ~60% (*Figure 3E*).

To summarize these behavioral changes, we used PCA to identify the best single axis of behavioral change across animals (PC1 explained 70.4% of variability; *Figure 3F*, see Materials and methods). We then fit a polynomial to the scores of control and gradually depleted mice (Adj-R$^2$ = 0.545; *Figure 3G*, see Materials and methods). The fit to gradually depleted mice was also a good predictor of symptomatic behavior in acutely depleted mice (*Figure 3G*, 'Acute'). The greater sensitivity of this analysis identified a trend towards modest hyperactivity in all tasks at intermediate stages of dopamine loss (~85–45%, *Figure 3F–G*). Thus, behavioral performance remained relatively intact until dopamine levels dropped below 25–35%, after which behavioral performance declined rapidly (*Figure 3G*). This sharp transition is similar to the pattern of symptomatic onset in human PD patients (*Bernheimer et al., 1973*; *Betarbet et al., 2002*; *Deumens et al., 2002*; *Fahn, 2003*; *Riederer and Wuketich, 1976*).

To determine whether changes in physiology (which decline biphasically with dopamine loss) are correlated with changes in behavior (which decline monophasically with dopamine loss), we fit a 2D polynomial to predict behavior PC1 using physiology PC1, and dopamine (*Figure 3H*, see: Materials and methods). Optimal behavioral prediction depended on physiology (y, y$^3$), dopamine (x, x$^2$), and their interaction (y*x$^2$) (95% confidence interval on coefficients, Adj-R$^2$ = 0.615, *Figure 3H*), suggesting physiology and behavior are correlated over the course of dopamine loss. To visualize the complex relationship between these variables, we overlaid the 2D fit of changes in

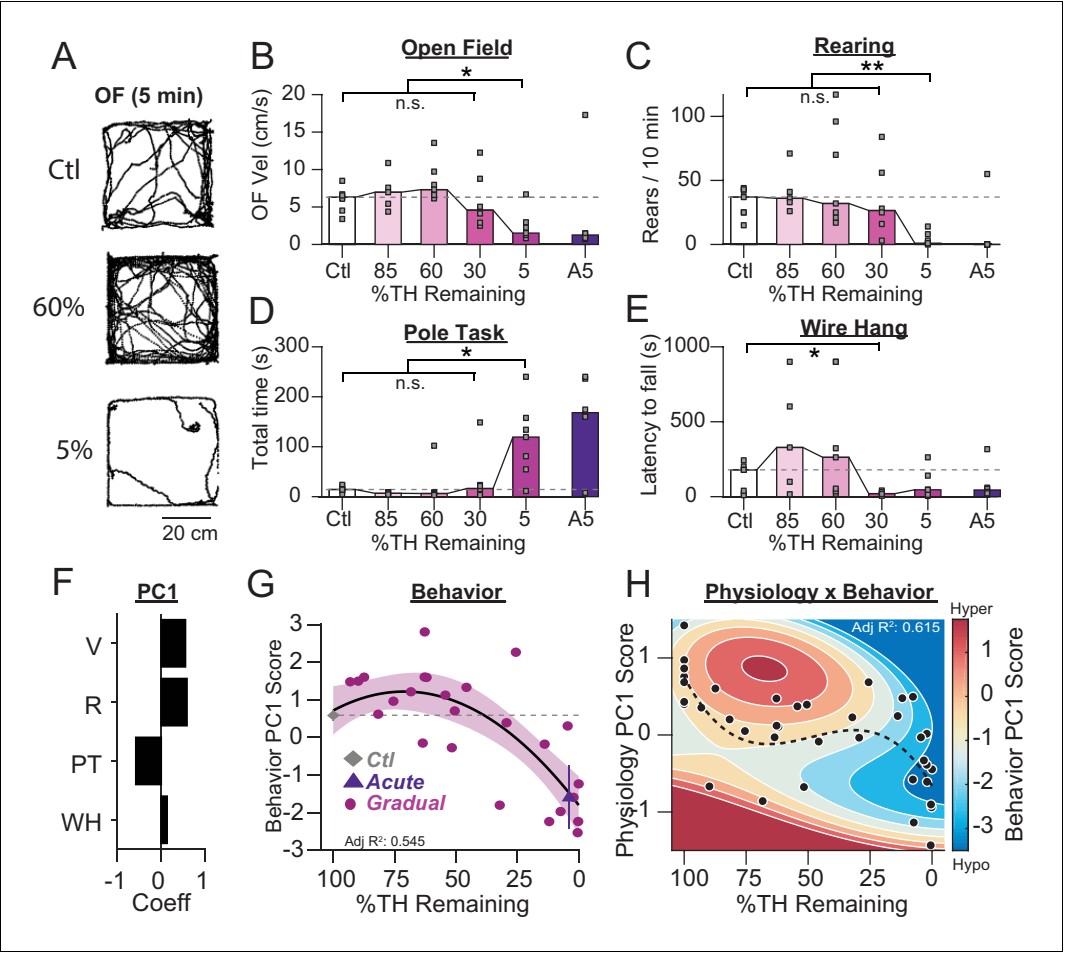

**Figure 3.** Motor deficits emerge at late stages of dopamine loss in 6-OHDA-treated mice. (A) Example raw open field movement traces from a Ctl, Gradual 60% and Gradual 5% mouse in a 5 min period. (B) Mouse open field velocity across depletion conditions. KW, $\chi^2(4) = 14.013$, $p = 0.007$, pairwise, $*p < 0.05$ from Ctl. (C) Rears per 10 min, across conditions. KW, $\chi^2(4) = 15.839$, $p = 0.003$, pairwise, $**p < 0.005$ from Ctl. (D) Total time completing a pole task. KW, $\chi^2(4) = 14.796$, $p = 0.005$, pairwise, $*p < 0.05$ from Ctl. (E) Wire hang duration across conditions. KW, $\chi^2(4) = 9.666$, $p = 0.046$. (F) 1st principal component coefficients for behavioral metrics. V, velocity; R, rearing; PT, pole task; WH, wire hang. (G) Mean animal behavior PC1 scores as a function of dopamine loss with model fit to Ctl and Gradual animals. Shaded region indicates $CI_{95}$ of fit (see Materials and methods). Average values (±SEM) from Ctl (grey diamond) and acutely depleted mice (purple triangle) are overlaid for reference. (H) 2D polynomial fit showing mouse behavior as a function of SNr pathophysiology and dopamine loss (n = 43 mice). Dashed line, physiology PC1 fit to Ctl, Gradual, and Acute mice. Adj. $R^2 = 0.415$.

DOI: https://doi.org/10.7554/eLife.42746.004

physiology +behavior + dopamine with the 1D fit of changes in physiology +dopamine (*Figure 3H*). This overlay reveals that early changes in physiology align with the prodromal period, while late changes in physiology align with the symptomatic period.

## PFF α-Syn drives gradual dopamine loss and behavioral changes that mirror the prodromal stage of 6-OHDA-treated mice

Our results in 6-OHDA treated mice reveal a hierarchical progression of physiological changes in the SNr over the course of gradual dopamine depletion. While most deficits emerged well before the onset of motor symptoms, dimensionality reduction exposed discrete physiological states that correlated with the prodromal and symptomatic stages of behavior. To determine whether the onset and progression of SNr pathophysiology is robust across depletion models, we transitioned from a toxin model to a neurodegenerative model.

The protein α-synuclein plays a central role in the pathogenesis of PD and is a major component of Lewy bodies in PD. A misfolded, fibrillar form of α-syn has been shown to propagate throughout the brain via cell-to-cell transmission and drive neurodegeneration of SNc dopamine neurons and the formation of Lewy body-like inclusions (*Luk et al., 2012*; *Mao et al., 2016*; *Volpicelli-Daley et al., 2014*). Striatal inoculation with preformed fibrils of α-syn (PFF α-syn) resulted in progressive dopamine loss and the formation of Lewy body-like inclusions (*Figure 4A–B*). Postmortem analysis of TH immunoreactivity in the striatum was used to group mice into discrete depletion categories that matched those of 6-OHDA-treated mice (TH immunoreactivity relative to control): 'Syn85%'=85 ± 10%; 'Syn60%'=60 ± 11%; 'Syn30%'=30 ± 5% (*Figure 4B*). Even after 6 months, striatal TH levels did not drop below 30%, thus end-stage physiological deficits could not be examined in the PFF α-syn model.

Mice with 85–30% dopamine remaining showed trends towards modest hyperactivity, similar to gradually depleted mice (*Figure 4C–F*), however the only group with significant differences from control was an increase in open field velocity in Syn60% mice. PFF α-Syn mice trended towards impaired wire hang performance but this difference was not significant (*Figure 4F*). We next calculated behavioral scores for each mouse from their performance across the battery of behavioral tasks and plotted them as a function of depletion stage (*Figure 4G–H*). Although PFF α-Syn mice did not reach end-stage depletion levels and therefore did not transition into the symptomatic stage, their behavioral scores were similar to that of 6-OHDA-treated mice at similar stages of depletion. This was most prevalent in the elevation of behavior scores at intermediate stages of depletion, driven by hyperactivity in the open field, more rearing, and decreased time on the pole task (*Figure 4H*, compare with *Figure 3G*).

## SNr pathophysiology progresses similarly across PFF α-Syn and 6-OHDA models

The onset and severity of SNr pathophysiology in PFF α-syn mice progressed along a similar trajectory as that seen in 6-OHDA-treated mice. SNr firing rates were significantly reduced even at the earliest stage of depletion, and this decrease persisted across subsequent stages of depletion (*Figure 5A*) (Median ±MAD: Control: 35 ± 11 Hz, $n$ = 262 neurons/7 animals; Syn85%: 26 ± 11 Hz, $n$ = 98/3; Syn60%: 25 ± 9 Hz, $n$ = 155/4; Syn30%: 25 ± 8 Hz, $n$ = 150/3; KW pairwise, Ctl vs. 85%, 60%, 30%, p < 0.0001).

Changes in firing patterns emerged at later stages of depletion (<60% remaining), mirroring the progression observed in 6-OHDA-treated mice. Firing irregularity ($CV_{isi}$) was increased (*Figure 5B*) (Median ±MAD: Ctl: 0.54 ± 0.13; Syn85%: 0.54 ± 0.10; Syn60%: 0.73 ± 0.19; Syn30%: 0.67 ± 0.22) as was the proportion of 'highly bursty' neurons (*Figure 5C*) (z-test, Ctl vs. Syn60%, Syn30%, p < 0.05).

At the population level, we saw a trend towards more synchronized spiking, but as was the case in 6-OHDA-treated mice, this effect never reached statistical significance during the prodromal period (*Figure 5D*) (Ctl: 8.6 ± 7.8% $n$ = 7 animals; Syn85%: 10.4 ± 2.6% $n$ = 3 animals; Syn60%: 13.7 ± 11.5 $n$ = 4 animals; Syn30%: 16.2 ± 19.4% $n$ = 3 animals; ANOVA, F(4) = 0.491, p = 0.742). Additionally, there was no increase in fractional β power measured at any stage of depletion in PFF α-syn mice (*Figure 5E*)(1-way ANOVA, p = 0.86).

To determine whether SNr pathophysiology progressed along a similar trajectory in PFF α-Syn mice compared to 6-OHDA-treated mice, we performed PCA (*Figure 5F–G*). In PFF α-Syn mice, PC1 captured changes in firing rate that were opposite in sign to irregularity, and bursting (51.8% variability explained; *Figure 5E*), similar to PC1 in gradually depleted mice (*Figure 2H*). In PFF α-Syn mice, PC1 showed a monophasic progression, presumably due to a lack of end-stage pathology (*Figure 5G*). This drop was similar to the initial drop seen in 6-OHDA treated mice, but a bit more gradual. Combined with results from 6-OHDA-treated mice, our data suggest that the progression of SNr pathophysiology depends more on the magnitude of dopamine depletion than the depletion model.

## Unilateral 6-OHDA depletion results in physiological changes in both hemispheres

Thus far, our study has examined the progression of SNr pathophysiology in bilaterally depleted mice. However in human PD, dopamine loss often begins asymmetrically (*Gelb et al., 1999*;

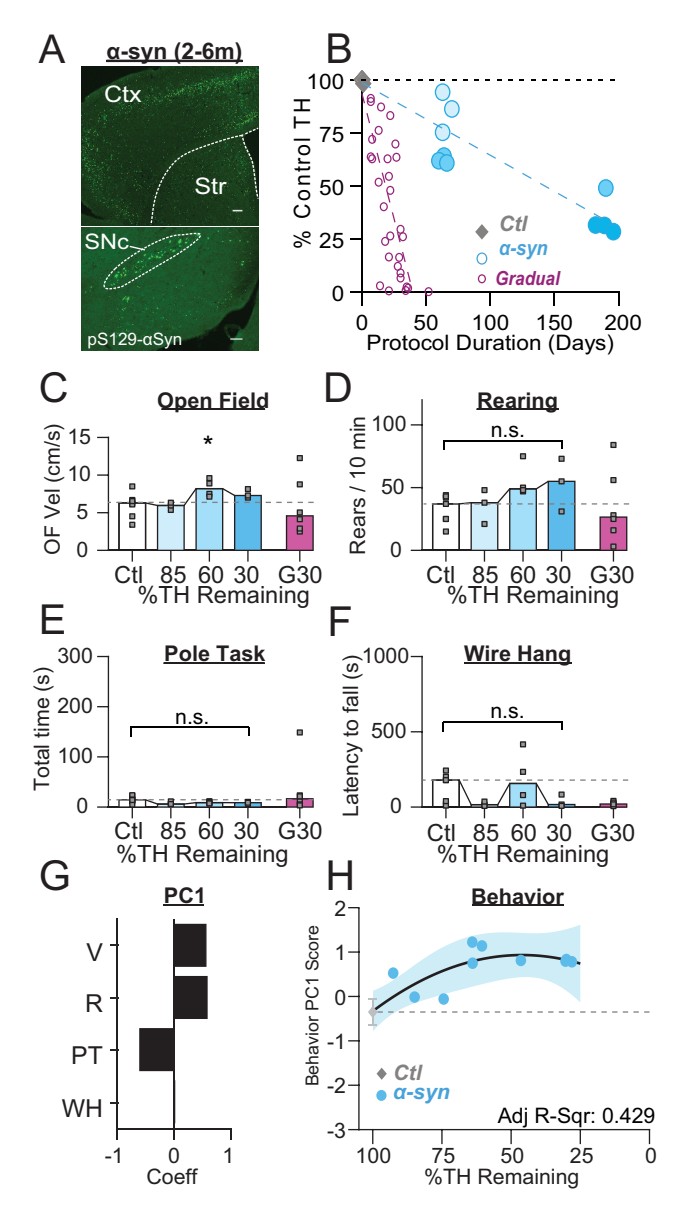

**Figure 4.** PFF α-Syn induces gradual dopamine depletion but fails to reach end stage. (**A**) Representative images of α-Syn inclusions present in the cortex (Ctx), striatum (Str), and substantia nigra pars compacta (SNc). Scale Bars = 100 μm (top), 200 μm (bottom). (**B**) Graph showing the trajectory of dopamine depletion induced by PFF α-syn (data from 6-OHDA-treated mice is replotted for reference). Percent TH remaining (relative to saline controls) is graphed as a function of protocol duration, measured from PFF α-syn infusion to day of sacrifice following in vivo recordings. (**C**) Mouse open field velocity across depletion conditions. ANOVA, $F(3) = 3.482$, $p = 0.047$, Dunnett, $*p < 0.05$ from Ctl. (**D**) Rears per 10 min, across conditions. KW, $\chi^2(3) = 7.540$, $p = 0.057$. (**E**) Total time completing a pole task. KW, $\chi^2(3) = 5.421$, $p = 0.143$. (**F**) Wire hang duration across conditions. ANOVA, $F(3) = 1.754$, $p = 0.205$. (**G**) 1st principal component coefficients for behavioral metrics. V, velocity; R, rearing; PT, pole task; WH, wire hang. (**H**) Mean animal behavior PC1 scores as a function of dopamine loss with model fit to Ctl and α-Syn animals. Shaded region indicates CI95 of fit.

DOI: https://doi.org/10.7554/eLife.42746.005

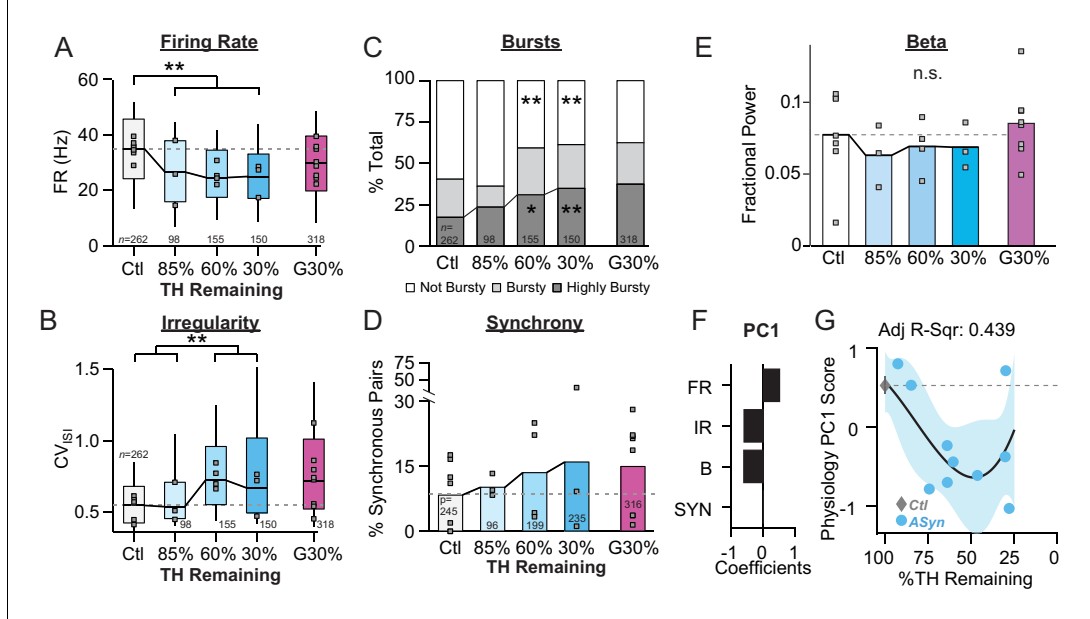

**Figure 5.** Onset and progression of SNr pathophysiology during gradual dopamine depletion in PFF α-Syn Mice. (A) Box plot of firing rates of all single units recorded at each stage of depletion. Grey squares indicate animal medians. KW, $\chi^2(4) = 53.274$, p < 0.0001, pairwise, **p < 0.005 from Ctl. Throughout the figure, data from G30% condition are re-plotted for reference. (B) Box plot of $CV_{ISI}$ of all single units recorded at each stage of depletion. Grey squares indicate animal medians. KW, $\chi^2(4) = 83.119$, p < 0.0001, pairwise, **p < 0.005 from Ctl and 85%. (C) Proportion of 'not bursty,' 'bursty,' and 'highly bursty' units at each stage of depletion. Pearson, $\chi^2(8) = 77.286$, p < 0.0001, z-test, *p < 0.05 and **p < 0.005 from Ctl. (D) Proportion of synchronous pairs at each stage of depletion (calculated as a percentage of total pairs sampled at that stage); grey squares indicate proportion of synchronous pairs for each mouse. ANOVA, F(4) = 0.491, p = 0.742. (E) Mean fractional β power (power in β relative to power 1–100 Hz) at each stage of depletion. Grey squares indicate animal means. ANOVA, p = 0.8564. (F) 1st principal component coefficients from PCA performed on Ctl and α-Syn single unit physiology: FR, firing rate; IR, irregularity; B, bursting; SYN, synchrony (see Materials and methods). (G) Mean animal physiology PC1 scores as a function of dopamine loss with model fit to Ctl and α-Syn animals. Shaded region indicates $CI_{95}$ of fit (see Materials and methods); Avg ± SEM for Ctl also shown.

DOI: https://doi.org/10.7554/eLife.42746.006

*Hoehn and Yahr, 1967*; *Hughes et al., 1992*), and the contralateral hemisphere might help to compensate (*Roedter et al., 2001*). Indeed, most studies of compensation under dopamine depleted conditions have been conducted in unilaterally depleted animals (*Chu et al., 2017*; *Escande et al., 2016*; *Fan et al., 2012a*; *Fieblinger et al., 2014*; *Fuller et al., 2014*; *Gittis et al., 2011*; *Taverna et al., 2008*), with suggestion that certain pathologies, such as β−oscillations, might require several weeks of unilateral depletion in order to fully manifest (*Brazhnik et al., 2014*; *Degos et al., 2009*; *Dejean et al., 2012*; *Leblois et al., 2007*; *Mallet et al., 2008*). To test whether unilateral depletion influences the nature or severity of pathophysiology in the SNr, we performed experiments in two unilateral models: 'unilateral' (*Figure 6A–B*) and 'asymmetric' (*Figure 6F–G*).

'Unilateral' mice received a single, high dose infusion of 6-OHDA (5 mg/mL) into the MFB of only one hemisphere, and SNr pathophysiology was measured 4–7 weeks later (39 ± 6 days), to match the final time point used for bilaterally depleted mice. Depletion severity was confirmed with TH immunoreactivity (*Figure 6B*) (1.6 ± 2.6% striatal TH remaining; n = 6 animals). A significant increase in TH immunoreactivity was observed on the contralateral side (211 ± 91%) (*Figure 6B*), suggestive of compensatory mechanisms engaged within the dopamine system (*Zigmond et al., 1990*; *Zigmond et al., 2002*).

Firing rates of SNr neurons in the depleted hemisphere were reduced relative to control (Median ±MAD: Ctl: 35 ± 11 Hz, n = 262 neurons/7 animals; Ipsi_uni: 26 ± 9 Hz, n = 136 neurons/6 animals; KW pairwise, Ctl vs. Ipsi_uni, p < 0.0001), and the magnitude of this change was similar to that seen in bilaterally depleted mice (*Figure 6C*) (KW pairwise, Ipsi_uni vs. G5%, p = 0.842). In five out of six unilaterally depleted mice, recordings were performed from the SNr in the contralateral hemisphere. Firing rates in the contralateral SNr were similar to those observed in control mice

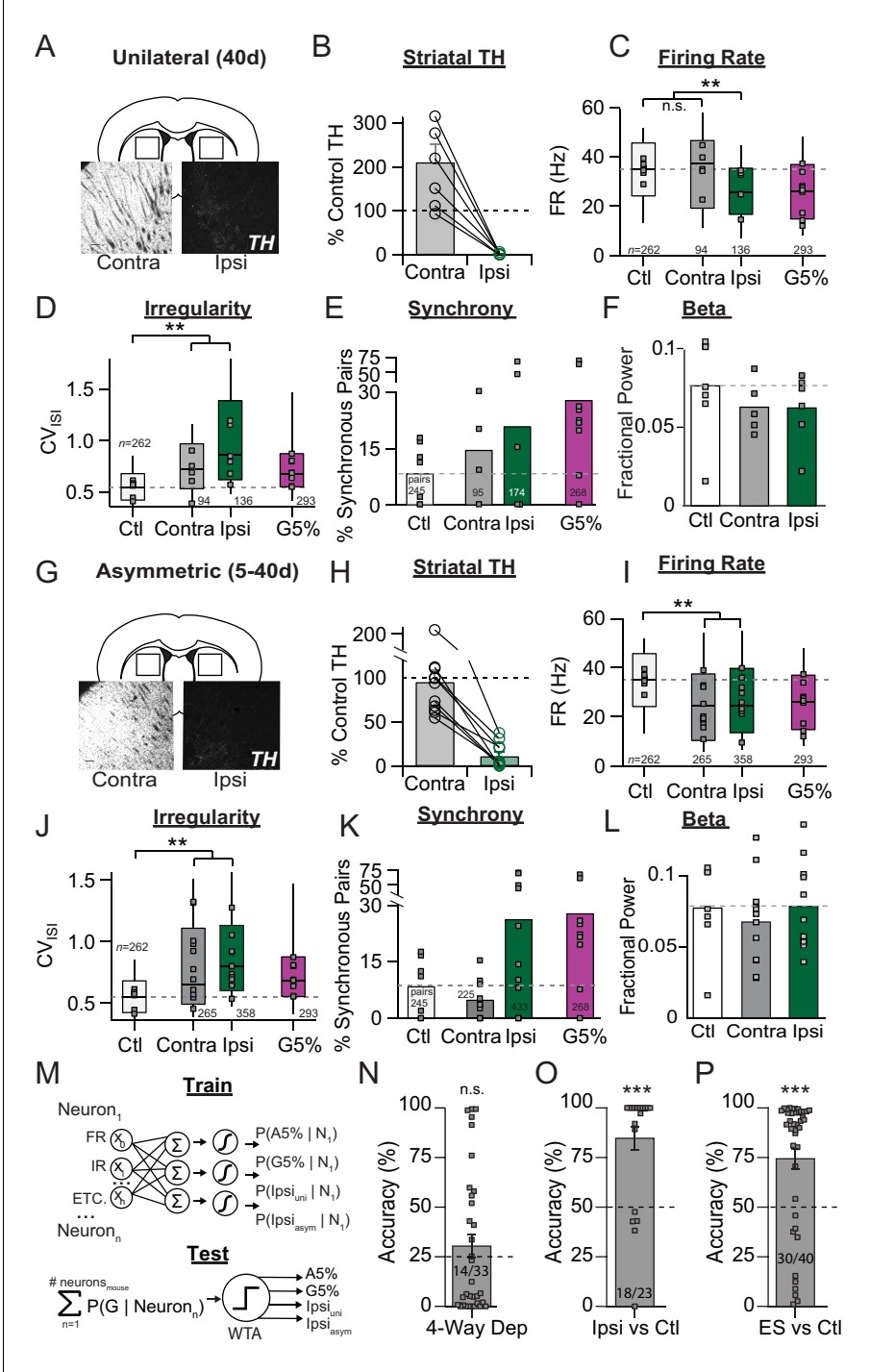

**Figure 6.** Unilateral 6-OHDA model pathophysiology aligns to prodromal and end-stage bilateral pathophysiology. (**A**) Example coronal sections showing TH-IR in the dorsal striatum of unilaterally depleted mice. Scale bar, 100 μm. (**B**) Quantification of the percent TH remaining (relative to control) in the ipsilateral ('Ipsi') and contralateral ('Contra') hemispheres of unilaterally depleted mice. (**C**) Box plots of firing rates from all neurons recorded in the ipsi and contra hemispheres. Grey squares indicate animal medians. KW, $\chi^2(3) = 5.488$, p < 0.0001, pairwise, **p < from Ctl. (**D**) Box plots of $CV_{ISI}$ from all neurons recorded in the ipsi or contra hemispheres. Grey squares indicate animal medians. KW, $\chi^2(3) = 110.685$, p < 0.0001, pairwise, **p < 0.005 from Ctl. (**E**) Proportion of synchronous pairs in ipsi or contra hemispheres (calculated as a percentage of total pairs sampled in each hemisphere); grey squares indicate proportion of synchronous pairs for each mouse. ANOVA, $F(3) = 1.297$, p = 0.300. (**F**) Mean fractional β power (power in β relative to power 1–100 Hz) at each stage of depletion. Grey

*Figure 6 continued on next page*

*Figure 6 continued*

squares indicate animal means. ANOVA, p = 0.4997. (**G-L**). Same as A-F but for asymmetrically depleted mice. I: KW, $\chi^2$(3) = 55.317, p < 0.0001, pairwise, **p < 0.005 from Ctl. J: KW, $\chi^2$(3) = 119.892, p < 0.0001, pairwise, **p < 0.005 from Ctl. K. KW, $\chi^2$(3) = 8.782, p = 0.032, *p < 0.05 from Ctl. L. ANOVA, p = 0.6660. (**M**) *Top*: Classifier trained to predict the probability that a neuron belongs to G5%, A5%, Ipsi$_{uni}$, Ipsi$_{asym}$ groups using: FR, firing rate; IR, irregularity; bursting, synchrony, and all pair-wise multiplicative interactions ('ETC.'). *Bottom*: predicted probabilities of group membership were summed across neurons from each held-out mouse (jack-knife), and thresholded using Winner-Take-All criterion (WTA). See: Materials and methods. (**N**) Average held-out mouse classification performance for G5%, A5%, Ipsi$_{uni}$ and Ipsi$_{asym}$ mice ('4-way class') using single unit physiology. Right-tail t-test vs. chance (25%), p = 0.1986, n = 33 mice. Grey squares indicate classification accuracy of individual held out mice. Inset text indicates mice predicted >chance/ total mice. (**O**) Same as M but for classifying Ctl from combined Ipsi$_{uni}$ and Ipsi$_{asym}$ mice ('Ipsi'). Right-tail t-test vs. chance (50%), p = 3.59 × 10$^{-6}$, n = 23 mice, ***p < 0.001. (**P**) Same as M but for classifying Ctl from all end-stage mice combined ('ES' = Ipsi$_{uni}$, Ipsi$_{asym}$, G5%, A5%). Right-tail t-test vs. chance (50%), p = 2.44 × 10$^{-5}$, n = 40 mice, ***p < 0.001.

DOI: https://doi.org/10.7554/eLife.42746.007

(*Figure 6C*) (Median ±MAD: Ctl: 35 ± 11 Hz, *n* = 262 neurons/7animals; Contra$_{uni}$: 37 ± 14 Hz, *n* = 94 neurons/5 animals) (KW pairwise, Ctl vs. Contra$_{uni}$, p = 0.712).

Changes in firing patterns were seen in both hemispheres. Firing irregularity (CV$_{isi}$) was significantly elevated on both sides, and this effect was stronger in the ipsilateral hemisphere compared to the contralateral hemisphere (*Figure 6D*) (Median ±MAD: Ctl: 0.54 ± 0.13 *n* = 262 neurons/7 animals; Ipsi$_{uni}$: 0.86 ± 0.30, *n* = 136 neurons/6 animals; Contra$_{uni}$: 0.72 ± 0.22, *n* = 94 neurons/5 animals; KW pairwise, Ctl vs. Ipsi$_{uni}$, Contra$_{uni}$, p < 0.0001). In both hemispheres, we saw a trend towards more synchronous spiking between pairs of neurons (Ctl: 8.6 ± 7.8% *n* = 7 animals; Ipsi$_{uni}$: 21.0 ± 28.2% *n* = 6 animals; Contra$_{uni}$: 14.8 ± 13.0% *n* = 5 animals), but this value did not reach significance (*Figure 6E*) (Dunnett, Ctl vs. Contra$_{uni}$, p = 0.930, Ctl vs. Ipsi$_{uni}$, p = 0.567). Finally, we looked at fractional β-power in unilaterally depleted mice, 4 weeks after depletion and found no significant difference (*Figure 6F*; 1-way ANOVA, p = 0.50).

Because the extreme dichotomy of the unilateral model is an exaggeration of the asymmetric dopamine loss observed in human PD (*Gelb et al., 1999*; *Hoehn and Yahr, 1967*; *Hughes et al., 1992*), we also looked at SNr pathophysiology in an 'asymmetric' model, in which 6-OHDA was infused bilaterally but produced asymmetric depletions, in which the difference in TH levels between the two hemispheres was >20% (Avg diff = 77 ± 40%, n = 11 animals). On average, TH immunoreactivity was 11 ± 14% in the ipsilateral hemisphere (Ipsi$_{asym}$) and 96 ± 44% in the contralateral hemisphere (Contra$_{asym}$) (*Figure 6G–H*).

In both hemispheres, firing rates of SNr neurons were significantly reduced compared to control (KW pairwise, Ctl vs. Ipsi$_{asym}$, Contra$_{asym}$, p < 0.0001), and the magnitude of this effect was similar to that in bilaterally depleted mice (*Figure 6I*). Firing irregularity (CV$_{isi}$) was also significantly elevated in both hemispheres (*Figure 6J*) (KW pairwise, Ctl vs. Contra$_{asym}$, Ipsi$_{asym}$, p < 0.0001). In contrast, firing synchrony was not equivalent between the two sides. There was a trend towards more synchronous pairs of neurons in the ipsilateral but not the contralateral hemisphere (*Figure 6K*). Lastly, no difference in fractional β-power was detected across these conditions (*Figure 6L*; 1-way ANOVA, p = 0.67).

To determine whether unilaterally depleted mice showed distinct physiological changes at end-stage compared to other depletion models, we trained a multinomial regression to classify mice into each of four end-stage categories (A5%, G5%, Ipsi$_{uni}$ and Ipsi$_{asym}$ conditions), using single unit physiology (*Figure 6M*)(see: Materials and methods). The classifier was unable to discriminate end-stage physiology above chance levels (30.4 ± 6.3% accuracy, right-tail t-test vs. chance = 25%; p = 0.1986; n = 33 mice) (*Figure 6N*). By contrast, combined Ipsi$_{uni}$ and Ipsi$_{asym}$ hemispheres were discriminated from Ctl mice with 84.7 ± 5.9% accuracy (*Figure 6O*) (right-tail t-test vs. chance, 50%; p = 3.59 × 10$^{-6}$, n = 23 mice). When combined into a single end-stage category, A5%, G5% and Ipsi depleted mice were well-discriminated from Ctl (*Figure 6P*) (74.4 ± 5.3% correct; right-tail t-test vs. chance, 50%; p = 2.44 × 10$^{-5}$, n = 40 mice). The majority of mice in each depletion model were correctly classified above chance (correct/total, A5%: 7/7 mice, G5%: 7/9 mice, Ipsi: 10/17 mice, Ctl:

6/7 mice). Thus, end-stage SNr physiology shows stereotyped pathophysiology across different depletion models.

*Physiological Changes Reveal State Transitions in the SNr that are Stereotyped Across Models*-Thus far, our analyses have established distinct physiological states in the SNr when mice are at early vs. late stages of dopamine loss. During this transition, the SNr appears to pass through an intermediate state, but it is unclear whether this represents a unique physiological state, or results from averaging early and late stages of physiology. To distinguish between these two possibilities, we first ran PCA on a data set compiled across all depletion models tested in the study. PC1 of this entire data set once again extracted changes in firing rate that were of opposite sign to changes in irregularity, bursting, and synchrony (*Figure 7—figure supplement 1A*), and followed a biphasic decline whose local minimum occurred midway through depletion (*Figure 7A* compared to *Figure 2I*).

To determine whether this biphasic decline was indicative of three discrete physiological states, we designed a multinomial classifier to test whether physiology could accurately predict the severity of depletion. To train the model, we segregated mice into three groups based on depletion severity ('early', intermediate', 'late'). The 'intermediate' group included mice whose dopamine levels fell within a symmetric window centered around the local minimum of PC1 (55 ± 20 %TH; n = 14 mice). The 'early' group included all mice above this window (100–75%, n = 15 mice), and the 'late' group included all mice below (35–0%, n = 41 mice). We then tested the classification accuracy of each mouse (jack-knife, 500 permutations), and whether it was predicted as 'early', 'intermediate' or 'late' depletion groups. The multinomial classifier successfully separated mice into each of these groups 52.9% of the time, significantly more often than expected by chance (33%) (*Figure 7C*) (multi-class Matthew's Correlation Coefficient = 0.25, p = 0.004, permutation test). Next, we trained a multinomial classifier to separate mice into four dopamine groups, however this classifier made errors consistent with there being only three groups (*Figure 7—figure supplement 1B–C*), further supporting three distinct physiological states. Finally, to test whether physiological states were stereotyped across depletion models, we reassessed 'early,' 'intermediate,' and 'late' mouse classification accuracy by instead grouping mice by depletion model (*Figure 7—figure supplement 1D*). We found that mice from all depletion models (Ctl, Acute, Gradual, α-Syn, and Ipsi), were classified into 'early,' 'intermediate,' and 'late' depletion groups above chance (p = 0.0237, $Chi^2$ = 11.3, Pearson's chi-square test). Taken together, these results suggest that over the course of dopamine loss, deterioration of SNr physiology occurs as a series of state transitions: from 'normal' to 'moderate', and from 'moderate' to 'severe'.

To determine which physiological changes were most responsible for each state transition, we plotted the multinomial coefficients used by the classifier to separate 'early,' 'intermediate,' and 'late' depletion groups (*Figure 7D*). These results suggest that transitions from the normal to moderate state were driven by increased irregularity, and firing rate interactions with spike patterns (*Figure 7D*). Transitions from the moderate to severe state were driven by increased synchrony (*Figure 7D*). To better visualize how changes in firing rates and patterns progress over the course of dopamine loss, and whether these changes promote or oppose dysfunction, we created a continuous measure of similarity between each parameter and its value at end-stage (*Figure 7E*, see: Materials and methods). During early dopamine loss, decreases in firing rate and increases in synchrony pushed the SNr towards pathology, transitioning from a 'normal' to 'moderate' pathophysiological state. As dopamine loss progressed, increases in irregularity and bursting promoted further dysfunction, but were offset by a decrease in synchrony and partial recovery of firing rate. These changes opposed dysfunction and possibly kept the SNr buffered in a 'moderate' pathophysiological state. At late stages of dopamine loss, however, this buffering broke down and dysfunction aligned across all physiological parameters, transitioning the SNr into a final, 'severe' pathophysiological state.

Finally, to capture the relationship between physiology and behavior across all bilateral depletion models, we fit a 2D polynomial to behavior PC1, physiological PC1, and dopamine (*Figure 7F*). Optimal behavioral prediction again depended on both physiology (y) and dopamine (x, $x^2$) (95% confidence interval on coefficients, Adj-$R^2$ = 0.57, *Figure 7F*, see: Materials and methods, compare to *Figure 3H*), suggesting that across depletion models, a similar correlation exists between physiology, behavior, and depletion severity. Taken together, these results expose the hierarchical progression of physiological changes in the SNr over the course of progressive dopamine loss and a stereotyped relationship between pathophysiological states and motor symptoms.

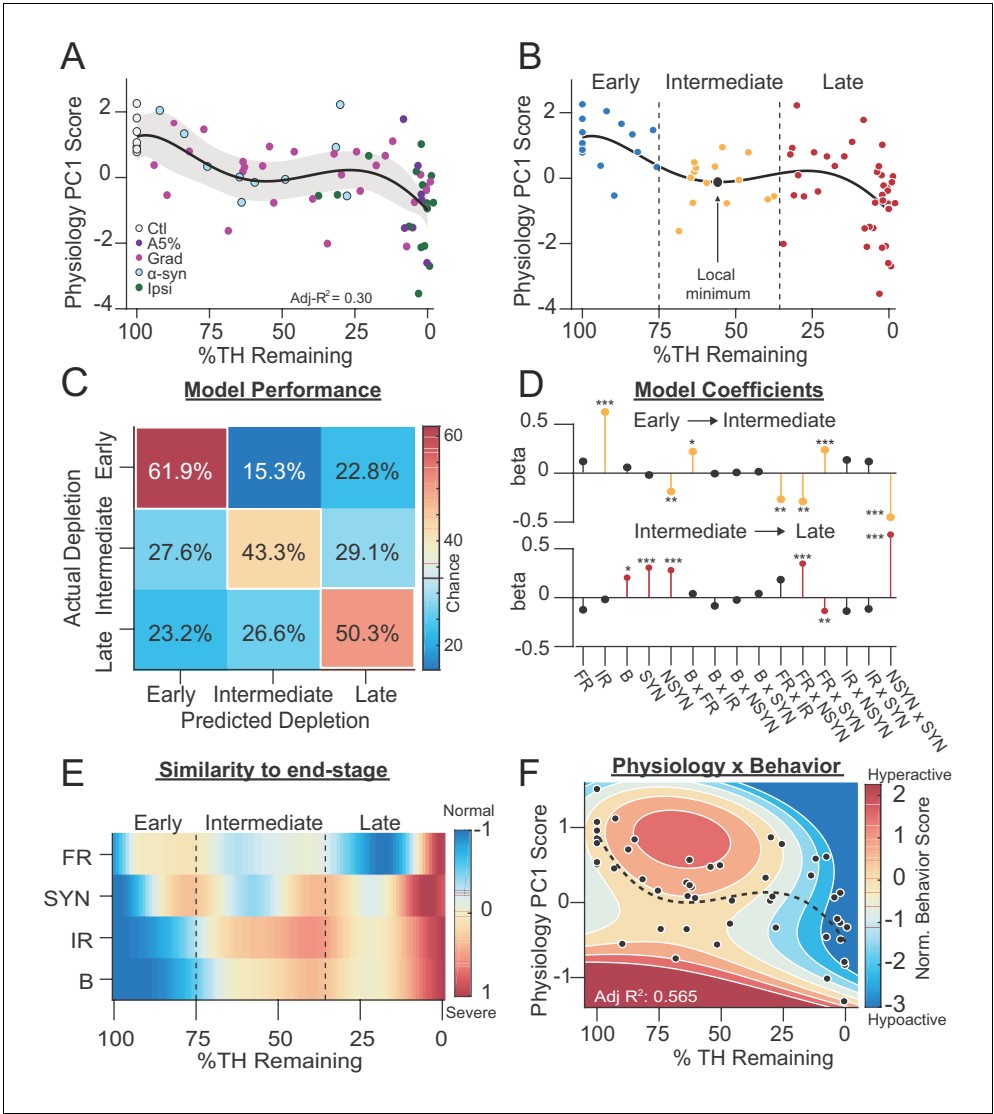

**Figure 7.** SNr exhibits distinct physiological states during progressive dopamine loss that are stereotyped across models. (**A**) Mean animal physiology PC1 scores as a function of dopamine loss with model fit to Ctl, Grad, PFF α-Syn, bilateral acute, Ipsi$_{asym}$, and Ipsi$_{uni}$ mice. Shaded region indicates CI$_{95}$ of fit (see Materials and methods; n = 68 mice). (**B**) Data from A with Early, Intermediate and Late dopamine groups defined around the local minimum at 55 %TH Remaining (±20%). (**C**) Confusion matrix of total cross-validated accuracy classifying mice into Early, Intermediate and Late dopamine groups using a multinomial regression (500 permutations, see: Materials and methods). Frequency of correct depletion severity predictions is highlighted along diagonal (chance = 33%). (**D**) *Top:* Average model coefficients for classifying Intermediate dopamine depletion relative to Early. *Bottom:* Average model coefficients for classifying Late dopamine depletion relative to Intermediate. (**E**) Instantaneous similarity of physiological parameters to end-stage via cross-correlation (n = 70 mice). FR, firing rate; SYN, percentage synchronous pairs per mouse; IR, irregularity (CV$_{ISI}$); B, percent spikes in bursts. (**F**) 2D polynomial fit showing mouse behavior as a function of SNr pathophysiology and dopamine loss in all bilateral conditions (n = 53 mice, Adj. R$^2$ = 0.565). Dashed line, physiology PC1 fit to all bilateral mice. Adj. R$^2$ = 0.27. See also: *Figure 7—figure supplement 1*.

DOI: https://doi.org/10.7554/eLife.42746.008
The following figure supplement is available for figure 7:

**Figure supplement 1.** Additional material corresponding to *Figure 7*.
DOI: https://doi.org/10.7554/eLife.42746.009

## Discussion

Our study reveals that over the course of progressive dopamine loss, SNr dysfunction progresses through a series of discrete state transitions. Small reductions in dopamine are sufficient to transition the SNr out of its 'normal' physiological state and into a 'moderate' pathophysiological state. The SNr remains buffered in this moderate pathophysiological state over a range of dopamine levels until finally transitioning into a 'severe' pathophysiological state at end-stages of depletion. These physiological transitions were well correlated with behavioral transitions into the prodromal and symptomatic states, respectively. The finding that SNr pathophysiology progresses in discrete stages, rather than smoothly as a function of dopamine loss, suggests different windows of opportunity for delivering specific therapies. Our results predict that interventions that preferentially affect firing rate (i.e. L-DOPA) are more likely to be effective at earlier stages of depletion rather than late, whereas interventions that preferentially affect regularity and synchrony (i.e. DBS) are more likely to be effective at late stages of depletion (*Figure 7D–E*). These results establish a conceptual framework for understanding the dynamic nature by which individual physiological parameters contribute to basal ganglia dysfunction over time and establishes a predictive model for therapeutic outcomes.

### Physiological states in the SNr predict behavioral states

Changes in firing rate, irregularity, bursts, and synchrony are well-established indicators of basal ganglia dysfunction under conditions of low dopamine (*Galvan and Wichmann, 2008*; *Obeso et al., 2000*; *Wichmann and Dostrovsky, 2011*), but their validity as biomarkers for different stages of PD has been underexplored. This is because most physiological studies are done at end-stages of depletion, or in symptomatic animals, precluding analysis of *when* physiological changes emerge, and whether they are accurate predictors of behavioral state. Previous attempts to link individual physiological parameters with motor symptoms and disease progression have yielded mixed results (*Bezard and Gross, 1998*; *Bezard et al., 2003*; *Leblois et al., 2007*; *Muralidharan et al., 2016*; *Stein and Bar-Gad, 2013*; *Tang et al., 2010*). For example, changes in firing rates are sufficient to transition animals between mobile and immobile states (*Kravitz et al., 2010*; *Lemos et al., 2016*), but during progressive dopamine loss, firing rate changes emerge before the onset of motor deficits (*Bezard et al., 1999*; *Bezard et al., 2001*; *Bezard et al., 2003*). Similarly, the therapeutic efficacy of DBS is well correlated with its ability to reduce synchrony (*Hammond et al., 2007b*; *Kühn et al., 2006*; *Ray et al., 2008*), but the degree of synchrony does not necessarily predict motor deficits (*Connolly et al., 2015*; *Leblois et al., 2007*; *Mallet et al., 2008*; *Muralidharan et al., 2016*). These and other observations have sparked controversy over whether changes in basal ganglia physiology underlie motor deficits in PD, or whether motor deficits are better predicted by physiological changes at areas outside the basal ganglia (*Bezard et al., 2003*; *Dirkx et al., 2017*; *Gradinaru et al., 2009*; *Wu and Hallett, 2013*).

Our results help to reconcile these seemingly contradictory results, by establishing the chronology of physiological changes in the SNr. Metrics of single-unit activity (rate, irregularity, bursts) were altered at early or intermediate stages of dopamine loss, but because these changes occurred out-of-phase with one another, they buffered the SNr in a 'moderate' pathophysiological state, during which motor function remained relatively intact (*Figure 7E–F*). Firing rates were the first parameter to change, followed by an increase in irregularity and bursting. Intriguingly, during the moderate pathophysiological state, modest hyperactivity was seen at the behavioral level in both 6-OHDA and PFF α-syn mice, perhaps reflecting the engagement of compensatory plasticity.

At late stages of dopamine depletion (<35% remaining), all physiological parameters began to change in unison, transitioning the SNr into a severe pathophysiological state (*Figure 7E–F*). This transition appeared to be driven by increased synchrony and correlated strongly with the onset of motor symptoms (*Figure 7D–F*). This result suggests a mechanism to account for a long-standing paradox of parkinsonian motor symptoms: alleviation of motor symptoms can be correlated with individual physiological parameters (*Hammond et al., 2007b*; *Kravitz et al., 2013*; *Kühn et al., 2006*; *Mastro et al., 2017a*; *Zhuang et al., 2018*), but rarely do single parameters predict the onset of motor symptoms (*Muralidharan et al., 2016*; *Nelson and Kreitzer, 2014*; *Sanders et al., 2013*). Our model reconciles this apparent paradox by predicting that motor symptoms emerge when the SNr is in the severe pathological state, which requires alignment of pathophysiology across many different parameters. Conversely, our data predict that changes in any individual parameter would be

sufficient to transition the SNr out of the severe pathological state and into the moderate pathophysiological state, where few if any motor deficits are observed. This prediction is supported by findings that Levodopa (L-DOPA) restores movement predominantly through effects on firing rates (*Aristieta et al., 2016*; *Hernandez et al., 2013*; *Parker et al., 2018*; *Ryan et al., 2018*) whereas DBS restores movement predominantly through effects on firing patterns (*Birdno and Grill, 2008*; *McCairn and Turner, 2009*; *McCairn et al., 2015*; *Zhuang et al., 2018*). We note, however, that the therapeutic mechanisms of these treatments remain an open question and could involve changes in both parameters.

## Comparison with previous studies

Most physiological changes observed in the SNr were consistent with previous literature, with two exceptions: amplified β-oscillations were not detectable in mouse SNr (see also *Lobb et al., 2013*), and SNr firing rates in our study were decreased by depletion, rather than increased, as predicted by the classical 'rate model' (*Albin et al., 1989*; *DeLong, 1990*). However, experimental measures of firing rate changes in the SNr following dopamine depletion have varied widely in their magnitude and direction across studies (*Seeger-Armbruster and von Ameln-Mayerhofer, 2013a*). One mechanism that might account for the firing rate decreases observed in our study is a decrease in current from type three canonical transient receptor potential channels (TRPC3) that is important for intrinsic pacemaking in the SNr (*Zhou, 2010*; *Zhou and Lee, 2011*; *Zhou et al., 2008*). TRPC3 channels are positively modulated by D1/5 receptors and their blockade has been shown to slow firing rates and promote irregular firing patterns.

Firing rate changes that take place in the SNr might also be different than those in the internal globus pallidus (GPi), the basal ganglia output nucleus that is preferentially targeted in human and primate studies. In a primate MPTP model, firing rate changes were more prominent in the GPi than in the SNr (*Wichmann et al., 1999*). Although rodents have a GPi-like structure, called the entopeduncular nucleus (EPN), it is much smaller than primate GPi, and only a subset of EPN neurons project to motor territories of the thalamus (*Kha et al., 2000*; *Wallace et al., 2017*)*Bezard and Gross, 1998*). By comparison, most neurons in rodent SNr encode motor information (*Barter et al., 2015*; *Rossi et al., 2016*) and can influence movement either through projections to motor thalamus or through projections to locomotor brainstem (*Capelli et al., 2017*; *Roseberry et al., 2016*).

## The role of compensatory plasticity

Compensatory plasticity is thought to play a major role in delaying the onset of motor symptoms until late stages of dopamine loss. Evidence of compensatory plasticity is seen throughout the basal ganglia and dopamine system including increased sensitivity to dopamine (*Bezard and Gross, 1998*; *Roedter et al., 2001*; *Schwarting and Huston, 1996*; *Zigmond et al., 1990*; *Zigmond et al., 2002*), synaptic and intrinsic changes within the striatum (*Blesa et al., 2012*; *Day et al., 2006*; *Escande et al., 2016*; *Fieblinger et al., 2014*; *Fuller et al., 2014*; *Gittis et al., 2011*; *Taverna et al., 2008*), and plasticity in the GPe/STN (*Chu et al., 2015*; *Chu et al., 2017*; *Fan et al., 2012a*). However, because most studies of compensatory plasticity have been performed at end-stages of dopamine loss, the extent to which these mechanisms offset or delay disease progression is poorly understood.

Our results suggest a role for compensatory plasticity in maintaining the intermediate physiological state in the SNr. At early stages of dopamine loss, SNr firing rates were initially decreased, but moved towards higher values as burst firing and irregularity became more severe (*Figure 7E*). Although we do not know the cellular mechanisms driving these physiological changes, they are consistent with hypotheses that bursting/irregularity emerge as a result of compensatory mechanisms engaged to counteract aberrant firing rates. In the striatum, firing rate changes in spiny projection neurons (SPNs) drive plasticity of inhibitory microcircuits that promote synchrony (*Gittis et al., 2011*). Synchronized output of the striatum can in turn promote irregular, burst/pause firing in the GPe (*Kita and Kita, 2011*), which further amplifies pathological synchrony throughout the circuit (*Corbit et al., 2016*; *Sharott et al., 2017*). Firing rate changes have also been shown to drive plasticity of connections between the GPe and STN, promoting oscillatory entrainment between these structures and contributing to pathological synchrony observed at later stages of depletion (*Chu et al., 2015*; *Chu et al., 2017*; *Fan et al., 2012b*).

With continued dopamine depletion, the physiological buffering observed during the intermediate state breaks down and the SNr transitions into a severe physiological state, coincident with the onset of motor symptoms. This severe physiological state was stereotyped across models, regardless of the time course or symmetry of depletions, suggesting that end-stage physiological deficits are not influenced by previous compensation. These results suggest that the transition from the asymptomatic to the symptomatic stage represents the tipping point at which the compensatory capabilities of the system become overwhelmed by the severity of dopamine depletion.

# Materials and methods

## Key resources table

| Reagent type (species) or resource | Designation | Source or reference | Identifiers | Additional information |
|---|---|---|---|---|
| Strain, strain background (*Mus musculus*) (M and F) | C57BL/6J mouse | Jackson Laboratory | stock_number: 000664; RRID: IMSR_JAX:000664 | Note: now bred in-house |
| Chemical compound, drug | 6-hydroxydopamine hydrobromide, 6-OHDA | Sigma Aldrich | stock_number: H116-5MG | (0.75 µg/hemisphere gradual; 5 µg/hemisphere - acute, unilateral) |
| Chemical compound, drug | 6-hydroxydopamine hydrobromide, 6-OHDA | Tocris | stock_number: 2547 CAS 636-00-0 | (5 µg/hemisphere - acute, unilateral) |
| Antibody | Anti-Tyrosine Hydroxylase, anti-TH rabbit | Pel-Freez Biologicals | RRID:AB_2617184; catalog_number: P40101-150; | (1:1000 diluted with glycerol) |
| Antibody | Alexa Fluor 647–conjugated donkey anti-rabbit | Life Technologies | RRID:AB_2536183; catalog_number: A-31573; | (1:500) |
| Antibody | anti-Iba-1 rabbit | Wako | RRID: AB_839504; catalog_number: 019–19741; lot_number: WDJ3047 | (1:1000) |
| Antibody | Rabbit Anti-Human alpha Synuclein, phospho (Ser129) Monoclonal; Unconjugated, Clone EP1536Y | AbCam | RRID:AB_869973; stock_number: ab51253 | (1:100) |
| Antibody | Alexa Fluor 488 donkey anti-rabbit | Life technologies | RRID:AB_2535792; catalog_number: A-21206; | (1:500) |
| Software, algorithm | SPSS Statistics | IBM | version:24 | |
| Software, algorithm | MATLAB | Mathworks | version:2018a | |
| Recombinant DNA reagent | recombinant a-syn | DOI: 10.1126/science.aah3374 | | (6 µg/hemisphere) |
| Software, algorithm | EthoVision XT 9.0 software | Noldus | RRID: SCR_000441 | |

Additional information and requests for reagents and resources will be fulfilled by the Lead Contact, Aryn Gittis (agittis@cmu.edu).

## Data availability

Processed data and code used to generate figure panels can be found online https://github.com/KidElectric/willard2018a.git (*Isett et al., 2019*; copy archived at https://github.com/elifesciences-publications/willard2018a).

## Animals

Experiments were conducted in accordance with the guidelines from the National Institutes of Health and with approval from Carnegie Mellon University Institutional Animal Care and Use Committee. Adult male and female mice (>90 days old) on a C57BL/6J background were used for experiments. After surgical implantation of the cannula or head-bar, animals were provided with dishes of crushed high fat food pellets moistened with water, additional hard food pellets on the floor of the cage, as well as access to a water bottle and all cages were placed half on/half off heating pads. For gradual 6-OHDA depletions, infusion of 6-OHDA were performed while animals were lightly anesthetized on a heating pad, and all animals were injected with 0.1 cc of saline i.p. before being returned to their home cage. Animal's weights were tracked regularly and extra i.p. saline and softened food or trail mix were provided to encourage weight gain and proper hydration when appropriate.

## Surgical procedures

*Cannula implantation:* Under ketamine/xylazine (100 mg/kg: 30 mg/kg, i.p.) anesthesia, the mice were placed on a stereotaxic frame (David Kopf Instruments) and maintained throughout surgery using 1–2% isoflurane. Bilateral internal cannulas (Plastics One) for delivery of 6-OHDA to the MFB were cut to target ±1.1 mm lateral and −5.0 mm ventral and were implanted 0.45 mm posterior to Bregma and secured using superglue. 6-OHDA was prepared at a concentration of 5 µg/µL in 0.9% NaCl for unilateral and acute bilateral depletions and diluted further with 0.9% NaCl to 0.75 µg/µL for bilateral gradual depletions (Sigma-Aldrich H116 6-Hydroxydopamine hydrobromide). Injections were performed using a 33-gauge cannula (Plastics One) attached to a 10 µL Hamilton syringe within a syringe pump (GenieTouch; Kent Scientific) running at 0.5 µL/min, to a total volume of 1 µL/side. The injection cannula was left in place for 5 min following the injection. For gradual depletions, 6-OHDA was administered every 5 days (for *n* days, depending on condition) and SNr recordings were performed 5 days after the last injection. In unilaterally depleted animals, SNr recordings were performed 4–7 weeks after 6-OHDA injections.

*Stereotaxic α-synuclein injection:* Animals undergoing the gradual PFF α-Syn paradigm underwent the same surgical preparation described above and received 1.5 µL injections of 4 µg/µL recombinant mouse α-synuclein pre-formed fibrils bilaterally into the striatum (AP:+0.5, ML: +/-, DV: −2.6 mm). Injections were performed as described above with an adjusted pump speed of 1.5 µL/7 min. PFF α-Syn animals were recorded from 2 to 6 months after injection.

## Immunohistochemistry

*TH immunoreactivity:* Degree of dopamine denervation was assessed in all animals based on immunofluorescence against tyrosine hydroxylase. Shortly after electrophysiological recordings, animals were sacrificed and perfused transcardially with phosphate-buffered saline (PBS), followed by 4% paraformaldehyde (PFA) in PBS. Brains were retrieved and post-fixed in 4% PFA for 24 hr before being rinsed with PBS, transferred to 30% sucrose in PBS, and stored at 4°C for at least 24 hr prior to sectioning. Immunohistochemistry was carried out in free-floating coronal frozen sections (30 µm). Tissue was sectioned using a freezing microtome (Microm HM 430; Thermo Scientific), blocked with 10% normal donkey serum, and permeabilized with 0.5% Triton X-100 for 1 hr. Primary antibody incubations were performed at room temperature for 24 hr using rabbit anti-TH (1:1000; Pel-Freez). Primary antibodies were detected with Alexa Fluor 647-conjugated donkey anti-rabbit (1:500, Thermo Fisher Life Technologies), incubated for 90 min at room temperature. Epifluorescent images (10x magnification) from TH staining were taken from bilateral dorsal striatum in one coronal section between 0.62 mm and 1.10 mm Bregma (according to Paxinos second edition Mouse Brain in Stereotaxic Coordinates). Pixel intensity over a $75 \times 75$ µm area (5625 µm$^2$) from each hemisphere was measured using the pixel intensity measuring tool in ImageJ and normalized to the pixel intensities measured in littermate control mice, processed and imaged in parallel.

*Iba-1 immunoreactivity:* Recording probe location was visualized with immunofluorescence against the microglial marker, Iba-1 (rabbit anti-Iba-1, 1:1000, Wako). Primary antibodies were detected with Alexa Fluor 647-conjugated donkey anti-rabbit (1:500, Thermo Fisher Life Technologies), incubated for 90 min at room temperature, or Alexa Fluor 488 donkey anti-rabbit (1:500, Thermo Fisher Life Technologies), incubated for 3 hr at room temperature.

*Plaque pathology*: Plaque pathology in PFF α-syn mice was assessed with phosphor-S129-α-synuclein staining (rabbit anti-phospho-S129-α-synuclein [EP1536Y], 1:100; AbCam). Primary antibodies were detected with Alexa Fluor 647-conjugated donkey anti-rabbit (1:500, Thermo Fisher Life Technologies), incubated for 90 min at room temperature, or Alexa Fluor 488 donkey anti-rabbit (1:500, Thermo Fisher Life Technologies), incubated for 3 hr at room temperature.

## Preparation of PFF α-synuclein

PFF α-syn were prepared by agitating recombinant α-syn with a magnetic stirrer (350 rpm at 37°C) for 7 days incubation. The aggregates were collected by the centrifugation (14,000 rpm 10 min) and were sonicated for 30 s at 10% amplitude (Branson Digital Sonifier, Danbury, CT, USA), and finally were aliquoted and kept at −80°C (*Mao et al., 2016*).

## Behavioral assessment

The day of in vivo recordings, animals were exposed to the following sequential behavioral tests before recordings: open field, rearing, pole task, and wire hang. The minimum interval between two consecutive procedures was 30 min. Mice were habituated to the testing room for 20 min before testing.

*Open Field:* To determine overall spontaneous mobility, mice were placed in the center of a 1,600 cm$^2$ clear square open field chamber with video monitoring from above. Mice were in the arena for a total of 20 m, with 10 m for acclimation to the arena, and 10 m for data acquisition. Positions of nose, tail, and center of mass of each mouse were tracked using EthoVision XT software (Noldus). Distance traveled and average velocities for the 10 m data acquisition period were calculated using EthoVision. The arena was cleaned with 50% ethanol in between animals.

*Rearing:* To assess spontaneous vertical activity, mice were placed in a standard 1000 mL glass beaker with video monitoring from the side for a total of 10 min. The number of full extension rears was manually scored post-hoc by observers blind to treatment. The beaker was cleaned with 50% ethanol between each animal.

*Pole Task:* To evaluate coordination and bradykinesia, mice were placed head-upward at the top of a vertical gauze-wrapped circular wooden pole (diameter = 1 cm; height = 55 cm) placed inside a clean home cage with video monitoring from the side. To encourage descent, a 60-watt lamp was aimed at the top of the pole. Mice were given a total of 6 trials, the first three for training, the last three for testing. The latency to turn downward (turn down latency = TDL), time from orientation downward until all four paws reached the ground (traverse), and the total time spent on the pole (total) was recorded with a maximum duration of 120 s each for TDL and traverse time. All measurements were manually scored offline by observers blind to treatment. Even if the mouse fell part way into its descent, the behavior was scored until it reached the ground. When the mouse was unable to turn downward and/or instead dropped from the pole, TDL and traverse latencies were recorded as 120 s (default value) because of the severity of motor dysfunction.

*Wire Hang:* Mice were placed on the top of a standard wire cage lid. The lid was slightly shaken to cause animals to grip the wires and then the lid was turned upside down and suspended ~50 cm above a standard animal cage with fresh bedding. The latency of mice to fall off the wire hang was measured up to 15 min, and average values were computed from two trials (15 min apart). Trials were stopped if the mouse remained on the lid after 15 min.

## In vivo SNr recordings

Head-bar implants to secure mice for in vivo recordings were performed under anesthesia as described above. Bilateral craniotomies (for probe insertion) were created over the SNr (−2.4 to −3.6 mm anterior, 0.9 to 2.1 mm lateral to Bregma) and a copper head-bar was fixed to the anterior portion of the skull (approximately at Bregma) using a combination of superglue and dental cement. Dental cement was extended from the head-bar to surround the extent of both craniotomies to

form a well. This well was then filled with silicone elastomer (Kwik-sil, WPI) that prevented infection and damage to the exposed brain tissue. During the recording, this well was filled with 0.9% NaCl and used as a ground reference. On the day of recording, animals were fixed to the top of the wheel and allowed 15 min to acclimate to the head-fixed position. The silicone elastomer was removed and the craniotomies were cleaned. A linear 16-channel silicon probe with sites spaced 50 μm apart (Neuronexus) was attached to the micromanipulator and centered on lambda. The probe was slowly advanced (5–7 μm/s) until the top of the SNr (~4.2 μm from the top of the brain) was found. SNr activity was distinguished based on a combination of physiological features: presence of putative dopamine neurons, presence of putative GABAergic neurons, and lack of spindle-like activity (thalamic). Post-mortem tissue analysis of Iba-1 (Wako) immunoreactivity induced by probe penetrations were further evidence of proper targeting. Once a population of SNr units was identified, 5–10 min of activity was recorded following a 5 min waiting period to ensure stability of the identified units.

## Electrophysiology analysis

Data was filtered at 150–8000 Hz for spiking activity and 0.7–300 Hz for local field potentials (LFP). Spike detection was completed using the Plexon offline sorter where principal component analysis was used to delineate single and multi-units. To be classified as a single unit, the following criteria will be utilized: (a) PCA clusters are significantly different (p<0.05); (b) J3-statistic is greater than 1; (c) percent of ISI violations (<1 ms) is less than 0.7%; (d) Davies Bouldin test statistic is less than 0.5. Following spike-sorting, data was processed with NeuroExplorer software in addition to custom scripts in MATLAB.

*Rest period analysis*: We measured rodent movement using an optical mouse with 10 Hz sampling and 0.03 mm spatial resolution. We identified movement onset when the average velocity in a 1.5 s window exceeded 0.2 mm/s, and expanded this window until the average within the expanded window fell below 0.1 mm/s. Any leading and lagging non-movement samples (samples where < 0.1 mm movement occurred) were excluded, and the resulting period was labeled as a 'movement bout.' Any period of time >0.5 s from a movement bout was labelled a 'rest bout.' Only rest bouts were used for firing rate, coefficient of variation, synchrony, burst and LFP analyses. Quantities calculated in each rest bout were combined into single values through an average weighted by the duration of each rest bout.

*LFP analysis:* Power spectral densities for each recording were calculated using the Lomb periodogram with values outside of bouts of rest removed. Beta power was defined as the total spectral power from 13 to 30 Hz. Fractional beta power was defined as the total spectral power from 13 to 30 Hz divided by the total power between 1–100 Hz.

*Burst analysis:* Using the Poisson Surprise method (surprise = 5), bursts were identified in single-unit SNr activity. 'Bursty units' were defined as units with >1% of total spikes occurring within a burst but less than one median absolute deviation above the median (i.e. 1–3.92% of spikes were in a burst), while 'highly bursty' units were defined as units with >1 median absolute deviation above the median of all recorded units (>3.92% of spikes were in a burst).

*Synchrony analysis:* We modified traditional cross-correlation analysis to correct for nonstationarities within a unit's firing pattern and to allow for direct comparisons across pairs of units regardless of their firing rates. We performed cross-correlation with a bin size of 10 milliseconds over 12-second-long windows with 4 s of overlap, excluding any window in which we detected movement on the running wheel. In each window, we zeroed the first and last 4 s of the 2nd train and only calculated out to a maximum lag of 4 s, thereby ensuring that each window would have a constant level of zero-padding across all calculated lags. This ensured a consistent level of baseline synchrony at long lags, which we used to normalize the cross-correlogram - specifically, we divided each window's cross-correlogram by the mean correlation value from 0.5 to 4 s on both sides. These normalized windows were each averaged together to achieve the final, normalized cross-correlation, whose values represent the proportion of synchronous spikes relative to the local chance level of synchrony (chance = 1). We calculated a 99% confidence interval from 0.5 to 4 s on both sides of the normalized cross-correlation and called a pair 'synchronous' if its normalized cross-correlation at zero lag exceeded this confidence interval. To calculate the fraction of synchronous pairs, we required that at least two units be recorded simultaneously, and the minimum number of simultaneously recorded pairs from an animal had to exceed 4. In addition, a single unit synchrony index was calculated by

taking the mean pair-wise normalized cross-correlation at zero lag for all simultaneously recorded units (used in *Figure 7A–E*).

## Quantification and statistical analysis

### Statistical analysis

All data sets were tested for normality with the Shapiro-Wilk test and equal variance with Levene's test prior to any statistical analysis. Data are expressed as median ±median absolute deviation (MAD) unless otherwise indicated. N values reported in text are formatted as follows: *n* = # of neurons/# of animals. Statistical analysis regarding firing rate and $CV_{ISI}$ was performed using Kruskal-Wallis analysis of variance (ANOVA) nonparametric test (KW) and any differences were further investigated by Kruskal-Wallis pairwise comparison between condition of interest and saline controls with a Bonferroni correction for number of comparisons. Statistical analysis regarding proportion of bursting units across conditions was performed using a Pearson Chi-Square Test (Pearson) and any further differences were investigated by z-test comparison of column proportions (z-test) between condition of interest and saline controls with a Bonferroni correction for number of comparisons. Statistical analysis regarding the average percentage of synchronous pairs was performed using a one-way ANOVA followed by a Dunnett t (2-sided) post hoc test with the exception of the asymmetric condition which was not normally distributed, thus we ran a KW test as described above. Results of initial statistical tests can be found in the figure legends, whereas any post-hoc testing is reported in results text where appropriate. A *p*-value of 0.05 was considered statistically significant. All statistical procedures were performed using IBM SPSS Statistics, version 24.

### Principal component analysis

*Physiology:* We performed centered, standardized PCA on single unit firing rate, $CV_{ISI}$, percent spikes in bursts, and % synchronous pairs per mouse (pca(), *Ryan et al., 2018*). In *Figure 7A,a* single unit synchrony index was included, as well as all multiplicative pair-wise interactions (*Figure 7—figure supplement 1A*). Coefficients were corrected for orthonormality. We applied a sign convention at the level of PC coefficients so that PC1 scores could be consistently interpreted relative to Controls. Next, we averaged single unit physiological PC scores within mice. Finally, we fit a polynomial model to mouse PC scores as a function of dopamine (see: Polynomial Fits). The best fit for PC1 was typically a 3$^{rd}$ or 5$^{th}$ degree polynomial (5$^{th}$ degree polynomial used in 7A). All units in each mouse (or hemisphere, in unilateral mice) were assigned the same % synchronous pairs value in this analysis. Skewed data ($CV_{ISI}$, percent spikes in bursts, synchrony measurements) were log-transformed prior to PCA (log10(), MATLAB).

*Behavior:* We performed centered, standardized PCA on mouse open field velocity, # of rears in 10 min, total time on pole task and wire hang latency (pca(), *Parker et al., 2018*). Coefficients were corrected for orthonormality We applied a sign convention at the level of PC coefficients so that PC1 scores could be consistently interpreted relative to Controls. Next, we fit a polynomial model to mouse behavioral PC scores as a function of dopamine (see: Polynomial Fits). The best fit for behavior PC1 was a 2$^{nd}$ degree polynomial based on Adj. $R^2$. Behavioral data were log-transformed prior to PCA to correct for skew (log10(), MATLAB).

### Polynomial fits

Polynomial fits were performed in *Ryan et al. (2018)* using linear least-squares regression (fit(), fittype = 'poly2', 'poly3','poly5,' for 1D fits, and 'poly23' for 2D fits). 95% confidence intervals of fit represent non-simultaneous bounds. Models were selected by optimizing Adj. $R^2$ values (reported in figures) and optimizing interpretability. Coefficient significance was evaluated using a 95% confidence intervals (see: fit(), *Zhuang et al., 2018*).

### Logistic and multinomial regressions for classification

To predict depletion state from single unit physiological features, we fit a cross-validated multinomial regression (mnrfit(), *Zhuang et al., 2018*). The model was trained to predict depletion state from unit firing rate, percent spikes in bursts, irregularity ($CV_{ISI}$), percentage of synchronous pairs per mouse, and all pair-wise multiplicative interactions of these parameters. For *Figure 7C–D,a* mean unit synchrony index was also included. We used a jackknife procedure for cross-validation: in

each iteration, one mouse was held out and the model was fit with the single unit data from all remaining mice (*Figure 1K* 'Train'). During this procedure, we created a balanced sample by resampling the number of single units to n = 50 and resampling the training mice to maintain an equal proportion of class examples (n = 500 permuted training sets and fits per held-out mouse). This allowed us to measure true performance against a chance performance of 100 * 1/n classes. To predict the held-out mouse's depletion state, we used the held-out single units as inputs to the fitted model, yielding a P(Depletion State | Physiology) for each neuron. We then summed these probabilities to determine the depletion state with the largest probability (Winner-Take-All threshold; *Figure 1K* 'Test'). Chance performance was verified by fitting a null model using scrambled mouse depletion state on each iteration. We measured whether depletion states could be discriminated above chance by performing a right-tailed t-test on the mean hold-out accuracy for each mouse compared to the mean accuracy of the null models for that same mouse. A multi-class Matthew's Correlation Coefficient was also calculated in multi-class comparisons showing significant t-tests. This value was compared to the null distribution of multi-class MCC values generated by the null models (described above) in order to create a p-value (n = 500 permutations).

## Instantaneous similarity to end-stage via cross-correlation

Ctl, bilateral 6-OHDA, Ipsi$_{asym}$, Ipsi$_{uni}$ and PFFα-Syn mouse physiology data were plotted as a function of dopamine loss and fit with a smoothing spline (fit(), SmoothingParam = 0.001, *Parker et al., 2018*). These fits were then padded, and cross-correlated with the pattern observed in the last 10% of TH loss (xcorr(), *Ryan et al., 2018*). This procedure was performed separately for each physiological parameter. Each cross-correlation was normalized so that data spanned from −1 to 1, representing least to most similar to end-stage.

## Acknowledgements

The authors would like to thank Jennifer Thomas (Seton Hill University) for her assistance with behavioral scoring and data analysis, and Ted M. and Valina L. Dawson's laboratories (John Hopkins University School of Medicine) for providing mouse α-synuclein pre-formed fibrils. This work was supported by National Institutes of Health Grants F31 NS093944 (AMW), F31 NS090745 (KJM), American Parkinson Disease Association (APDA) Research Grant (XM) NIH/National Institute on Aging Grant 1K01AG056841-01 (XM), R21 NS095103 (AHG), and R01 NS101016-01 (AHG).

## Additional information

### Funding

| Funder | Grant reference number | Author |
| --- | --- | --- |
| NIH Office of the Director | NS095103 | Aryn H Gittis |
| NIH Office of the Director | NS101016-01 | Aryn H Gittis |
| NIH Office of the Director | NS093944 | Amanda M Willard |
| NIH Office of the Director | NS090745 | Kevin J Mastro |
| NIH Office of the Director | NS101821 | Timothy C Whalen |
| NIH Office of the Director | K01AG056841-01 | Xiaobo Mao |
| American Parkinson Disease Association | | Xiaobo Mao |

The funders had no role in study design, data collection and interpretation, or the decision to submit the work for publication.

### Author contributions

Amanda M Willard, Conceptualization, Data curation, Formal analysis, Funding acquisition, Investigation, Methodology, Writing—original draft, Writing—review and editing; Brian R Isett, Conceptualization, Data curation, Software, Formal analysis, Validation, Investigation, Visualization,

Methodology, Writing—original draft, Writing—review and editing; Timothy C Whalen, Formal analysis, Methodology; Kevin J Mastro, Data curation; Chris S Ki, Formal analysis; Xiaobo Mao, Resources, Methodology; Aryn H Gittis, Conceptualization, Formal analysis, Supervision, Funding acquisition, Validation, Writing—original draft, Project administration, Writing—review and editing

## Author ORCIDs
Amanda M Willard (ID) https://orcid.org/0000-0003-3485-252X
Brian R Isett (ID) https://orcid.org/0000-0002-1581-0706
Timothy C Whalen (ID) https://orcid.org/0000-0002-1122-1090
Aryn H Gittis (ID) http://orcid.org/0000-0002-3591-5775

## Ethics
Animal experimentation: Experiments were conducted in accordance with the guidelines from the National Institutes of Health and with approval from Carnegie Mellon University Institutional Animal Care and Use Committee (protocol # AS15-018).

## Decision letter and Author response
Decision letter https://doi.org/10.7554/eLife.42746.012
Author response https://doi.org/10.7554/eLife.42746.013

## Additional files

### Supplementary files
• Transparent reporting form
DOI: https://doi.org/10.7554/eLife.42746.010

### Data availability
All data generated or analysed during this study are included in the manuscript and supporting files found on https://github.com/KidElectric/willard2018a (copy archived at https://github.com/elifesciences-publications/willard2018a).

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
