## [Decision Letter]

Thank you for submitting your article "State Transitions in the SNr Predict the Onset of Motor Deficits in Models of Progressive Dopamine Depletion in Mice" for consideration by *eLife*. Your article has been reviewed by Eve Marder as the Senior Editor, a Reviewing Editor, and three reviewers. The reviewers have opted to remain anonymous.

The reviewers have discussed the reviews with one another and the Reviewing Editor has drafted this decision to help you prepare a revised submission.

Summary:

Parkinson's disease (PD) results from loss of dopaminergic neurons in the SNc. While previous studies of PD have identified alterations in key physiological parameters in the basal ganglia output nuclei (SNr), analyses have been largely limited to the end stage of the disease. Here, Willard et al. use different dopamine depletion mouse models, in which striatal dopamine is depleted acutely or gradually by 6-OHDA lesions (unilaterally or bilaterally) or by α-synuclein fibril injection; they characterize physiology and motor behavior as a function of the severity of dopamine depletion across models and apply a host of computational approaches to analyze data. Based on the observations, it is concluded that there are discrete state transitions in physiology that track the progression of dopamine depletion, such as the modest changes in both the rate and pattern (irregularity and synchrony) of SNr firing that accompany the initial dopamine loss. Notably, the state transitions associated with the degree of dopamine loss seem largely independent of the type of model used or the rate of progression. The experiments are elegant, and although the findings are largely descriptive, the main results are likely to be valuable in the PD research field. Furthermore, the use of computational approaches is an enticing direction. Altogether the reviewers find the study to be impactful and are enthusiastic about the manuscript. However, the following concerns regarding analyses and interpretations require careful consideration.

Essential revisions:

1) It is of concern that the authors over-rely on fitting the PCA trajectory to a polynomial line and using this fit to claim a biphasic progression. Fitting polynomials to these kinds of data seems rather risky because a few "outliers" could dramatically affect the shape of the curve. While the concept of applying PCA on various physiological parameters is attractive, it is not clear if these polynomial fits are sufficiently strong evidence of biphasic progression or distinct states. Relatedly, subsection “Physiological Changes Reveal State Transitions in the SNr that are Stereotyped Across Models” states "three distinct physiological states…" Figure 7 would be much more supportive of this claim if it used the multinomial classifier the authors developed to try to classify the% TH remaining from the physiology. Figure 7E and 7G partly go in that direction but it's not enough, because they only show that all animals with end-stage TH depletion have similar physiology. To prove that there are at least three distinct states, the authors should split the classifier into at least three TH groups (such as high, medium, low). Another suggestion is to plot PC1 vs. PC2 for the high, medium, and low TH groups on the same graph, and hopefully show that each group tends to be clustered in PC space. This may provide a clearer proof of physiological state transitions than the polynomial fits.

2) Figure 7 is quite confusing in a few ways. For example, it's unclear what's being plotted in 7B. The caption reads "normalized bilateral physiology PC1 and behavior PC1." But the figure only appears to show physiology PC1. Assuming the "behavior PC1" is a misprint, it is not obvious how the PC1 curves for Ipsi and Contra end up as perfect lines with zero slope. Again, the revision should not use polynomial fits as the main supporting evidence for state transitions.

3) Subsection “Gradual Dopamine Depletion With 6-OHDA Results in Late Onset of Behavioral Deficits”: Figure 2E doesn't do a very good job of showing whether SNr physiology is correlated with behavior. A better (and probably simpler) approach would be to plot the physiology PC1 vs. the behavior PC1, or physiology PC2 vs. behavior PC1, and check for statistically significant correlations, or other interesting trends. Similar comment for Figure 5I, 5J.

4) Subsection “SNr Pathophysiology Progresses Similarly Across PFF α-Syn and 6-OHDA Models” and conclusion in general. Although it is stated that "the progression of SNr pathophysiology depends more on the stage of dopamine depletion than the mechanism of depletion", the severity of progression itself is defined by the state of dopamine depletion, and thus the argument is circular. The authors should consider rephrasing the statement to avoid confusion.

---

## [Author Response]

Essential revisions:1) It is of concern that the authors over-rely on fitting the PCA trajectory to a polynomial line and using this fit to claim a biphasic progression. Fitting polynomials to these kinds of data seems rather risky because a few "outliers" could dramatically affect the shape of the curve. While the concept of applying PCA on various physiological parameters is attractive, it is not clear if these polynomial fits are sufficiently strong evidence of biphasic progression or distinct states. Relatedly, subsection “Physiological Changes Reveal State Transitions in the SNr that are Stereotyped Across Models” states "three distinct physiological states…" Figure 7 would be much more supportive of this claim if it used the multinomial classifier the authors developed to try to classify the% TH remaining from the physiology. Figure 7E and 7G partly go in that direction but it's not enough, because they only show that all animals with end-stage TH depletion have similar physiology. To prove that there are at least three distinct states, the authors should split the classifier into at least three TH groups (such as high, medium, low). Another suggestion is to plot PC1 vs. PC2 for the high, medium, and low TH groups on the same graph, and hopefully show that each group tends to be clustered in PC space. This may provide a clearer proof of physiological state transitions than the polynomial fits.

We agree that evidence of three physiological states would be made more compelling if supported by a second, independent approach. As suggested by reviewers, we trained a multinomial classifier to separate mice into Early (100-75%), Intermediate (75-35%) and Late (35-0%) DA groups based on SNr physiology (Figure 7B-D). This approach successfully classified mice into each group. We then trained a multinomial classifier to separate mice into four DA groups, however this classifier made errors consistent with there being only 3 groups (Figure 7—figure supplement 1B-C). We interpret these results to support the hypothesis that there are three distinct physiological states.

2) Figure 7 is quite confusing in a few ways. For example, it's unclear what's being plotted in 7B. The caption reads "normalized bilateral physiology PC1 and behavior PC1." But the figure only appears to show physiology PC1. Assuming the "behavior PC1" is a misprint, it is not obvious how the PC1 curves for Ipsi and Contra end up as perfect lines with zero slope. Again, the revision should not use polynomial fits as the main supporting evidence for state transitions.

This figure has been removed. Data to support the conclusion that unilaterally depleted mice exhibit the same physiological states as bilaterally depleted mice are provided: (1) in Figure 6M-O we show that a multinomial classifier does not discriminate the depletion method used to reach end-stage dopamine levels, and (2) in Figure 7A we show that the PC1 scores of unilaterally depleted mice follow the same trend as bilaterally depleted mice.

3) Subsection “Gradual Dopamine Depletion With 6-OHDA Results in Late Onset of Behavioral Deficits”: Figure 2E doesn't do a very good job of showing whether SNr physiology is correlated with behavior. A better (and probably simpler) approach would be to plot the physiology PC1 vs. the behavior PC1, or physiology PC2 vs. behavior PC1, and check for statistically significant correlations, or other interesting trends. Similar comment for Figure 5I, 5J.

We agree with the criticism here and worked to improve the analysis. First, we attempted the reviewers’ suggestion, but realized that plotting physiology vs. behavior was contaminated by covariation with dopamine, making the relationship difficult to interpret. Thus, we decided to show how dopamine and physiology predict mouse behavior when all three are fit in a 2D model. We now show the resulting fit of this model as a contour plot in Figure 3H, and again for all mice in Figure 7F.

4) Subsection “SNr Pathophysiology Progresses Similarly Across PFF α-Syn and 6-OHDA Models” and conclusion in general. Although it is stated that "the progression of SNr pathophysiology depends more on the stage of dopamine depletion than the mechanism of depletion", the severity of progression itself is defined by the state of dopamine depletion, and thus the argument is circular. The authors should consider rephrasing the statement to avoid confusion.

The reviewers’ point is well taken. We now rephrase this statement as: “The progression of SNr pathophysiology depends more on the magnitude of dopamine depletion than the depletion model.”